# Fair Decision Utility in Human-AI Collaboration: Interpretable Confidence Adjustment for Humans with Cognitive Disparities

**Jiashi Gao**[1]   **Kexin Liu**[1]   **Xinwei Guo**[1]   **Junlei Zhou**[1]
**Jiaxin Zhang**[1]   **Xiangyu Zhao**[2]   **Guanhua Chen**[1]   **Xin Yao**[3]   **Xuetao Wei**[1*]

[1]Southern University of Science and Technology
[2]City University of Hong Kong    [3]Lingnan University

## Abstract

In AI-assisted decision-making, human decision-makers finalize decisions by taking into account both their human confidence and AI confidence regarding specific outcomes. In practice, they often exhibit heterogeneous cognitive capacities, causing their confidence to deviate, sometimes significantly, from the actual label likelihood. We theoretically demonstrate that existing AI confidence adjustment objectives, such as *calibration* and *human-alignment*, are insufficient to ensure fair utility across groups of decision-makers with varying cognitive capacities. Such unfairness may raise concerns about social welfare and may erode human trust in AI systems. To address this issue, we introduce a new concept in AI confidence adjustment: *inter-group-alignment*. By theoretically bounding the utility disparity between human decision-maker groups as a function of *human-alignment* level and *inter-group-alignment* level, we establish an interpretable fairness-aware objective for AI confidence adjustment. Our analysis suggests that achieving utility fairness in AI-assisted decision-making requires both *human-alignment* and inter-group-alignment. Building on these objectives, we propose a multicalibration-based AI confidence adjustment approach tailored to scenarios involving human decision-makers with heterogeneous cognitive capacities. We further provide theoretical justification showing that our method constitutes a sufficient condition for achieving both *human-alignment* and *inter-group-alignment*. We validate our theoretical findings through extensive experiments on four real-world tasks. The results demonstrate that AI confidence adjusted toward both *human-alignment* and *inter-group-alignment* significantly improves utility fairness across human decision-maker groups, without sacrificing overall utility. *The implementation code is available at* https://github.com/WEILaboratory/AI-Ethics-Safety-PaperCode/tree/main/Fair_HAI (ICLR2026).

## 1 Introduction

In recent years, artificial intelligence (AI) has been increasingly leveraged to assist human in decision-making across various domains. For example in typical binary classification tasks, AI systems have been developed to support clinicians in medical diagnosis (Rajpurkar et al., 2020; Wysocki et al., 2023), aid financial institutions in credit risk assessment (Bussmann et al., 2021), and assist legal professionals in bail or sentencing judgments (Dement & Inglis, 2024; Grgić-Hlača et al., 2019). However, AI systems trained on datasets inherently exhibit uncertainties and remain far from achieving accurate predictions in every case (Prabhudesai et al., 2023); a recommended approach under uncertainty considerations is to have AI systems provide confidence ranging from 0 to 1 for their outputs, which enables human decision-makers to better interpret and utilize the AI's assistance in their decision-making processes (Bhatt et al., 2021; Steyvers & Kumar, 2024; Ma et al., 2023). Consequently, the AI-assisted decision-making process is structured as follows: human decision-makers integrate their own confidence on specific labels with AI's to make final decisions.

---

*Corresponding author: weixt@sustech.edu.cn

Ideally, human decision-makers place greater reliance on AI when its confidence is higher and rely more on their own judgment when the AI's confidence is lower.

Existing research in AI-assisted decision-making has primarily focused on enhancing final decision-making's utility (i.e., accuracy in classification tasks or prediction errors in regression tasks). Early studies suggested that AI confidence should be *well-calibrated* estimates of the probability that the predicted label matches the truth label (Pakdaman Naeini et al., 2015; Yin et al., 2019; Zhang et al., 2020). However, Vodrahalli et al. (2022b) experimentally demonstrated that in certain scenarios, explicitly uncalibrated AI advice led to substantially higher decision utility compared to well-calibrated advice above. Subsequently, Corvelo Benz & Rodriguez (2023) provided a detailed theoretical analysis, demonstrating that rational decision-makers make optimal final decisions when AI confidence exhibits a natural alignment with human decision-makers' confidence in their own predictions, referred to as *human-alignment*.

Human decision-makers, shaped by different historical and social contexts, exhibit varying cognitive capacities on decision-making tasks, causing their confidence to deviate by different magnitudes from the actual likelihood of the labels. We demonstrate that existing AI confidence calibration mechanisms fail to provide fair decision utility across diverse groups of human decision-makers, who are differentiated by their cognitive capacities. Such unfairness may undermine human decision-makers' trust and willingness to engage with AI. Furthermore, it may exacerbate existing disparities in access to information and knowledge. For example, in AI-assisted medical diagnosis, unequal AI support might widen the diagnostic error gap between less experienced doctors (who typically have lower cognitive capacities) and experts (who typically have higher cognitive capacities). Such AI-assisted decision-making system not only limits opportunities for skill improvement among less experienced professionals but also risks reinforcing the *Matthew Effect* (Merton, 1968)—a socioeconomic phenomenon whereby disparities in access to resources (e.g., education, economics, or information) become self-reinforcing over time, leaving already disadvantaged groups even further marginalized. **Therefore, in this work, we aim to mitigate the utility disparities in AI-assisted decision-making that arise from human decision-makers' heterogeneous cognitive capacities.**

**Our contributions.** To the best of our knowledge, this is the first work to identify and address the utility fairness issue affecting human decision-makers in AI-assisted decision-making, and we make the detailed contribution as follows: ① We identify a previously unexplored source of utility unfairness in human-AI collaboration, which arises from human decision-makers with heterogeneous cognitive capacities. We theoretically demonstrate that existing AI confidence adjustment objectives, including *calibration* and *human-alignment*, are insufficient to guarantee fair utility for such heterogeneous human decision-makers. ② We introduce a novel AI confidence adjustment objective, *inter-group-alignment*, and derive a theoretical upper bound on utility disparity that is determined by the degrees of both *inter-group-alignment* and *human-alignment*. This provides actionable insight into how AI confidence should be configured to achieve fair decision utility. ③ We provide a theoretically grounded unification of the dual objectives, *inter-group-alignment* and *human-alignment*, under a multicalibration-based AI confidence adjustment approach. This approach offers a practical solution for achieving both utility fairness and optimal utility among human decision-makers with heterogeneous cognitive capacities in AI-assisted decision-making. ④ We conduct extensive experiments on four real-world AI-assisted decision-making tasks involving actual human decision-makers. The results validate our theoretical findings and demonstrate that adjusting AI confidence toward both *inter-group-alignment* and *human-alignment* substantially improves utility fairness across groups of human decision-makers, without compromising overall utility.

## 2 PRELIMINARY

We focus on binary decision-making scenarios to investigate the presence of unfair utility among human decision-maker groups with diverse cognitive capacities in existing AI confidence calibrations and to explore potential solutions. Such scenarios are common in real-world applications, including loan approvals, disease diagnoses, and job assignments. We define the attributes used to group the human decision-makers as $S \in \mathcal{S}$. The variable $S$ can represent any human characteristic that may contribute to disparities in cognitive capacities for a given task, including direct features (e.g., level of education) or indirect features (e.g., gender, race), which may have historically resulted in unequal access to knowledge resources. Let $f_H : \mathcal{X} \to [0, 1]$ denote the human decision-maker's confidence

function for positive outcomes. Initially, the human decision-maker observes a sample with features $x \in \mathcal{X}$ and assigns a confidence $h = f_H(x) \in \mathcal{H}$. Subsequently, the AI model (i.e., a classifier) provides its confidence value $a = f_A(x, h, s)$, where $s$ represents a specific value of the variable $S$, indicating the group to which the human decision-maker belongs. Here, $f_A : \mathcal{Z} \rightarrow [0, 1]$ denotes the AI's confidence function for positive outcomes, with $\mathcal{Z} = \{\mathcal{X}, \mathcal{H}, \mathcal{S}\}$. Finally, the human decision-maker makes a binary decision $T$ based on the probability $P(T = 1) = \pi(h, a) \in \{0, 1\}$. Specifically, $T = 1$ if $P(T = 1) \geq 0.5$, and $T = 0$ otherwise. Here, $\pi \in \Pi(\mathcal{H}, \mathcal{A})$ denotes the decision-making policy. Upon making a decision, the decision-maker receives a utility $u(T, Y) \in \mathbb{R}$ under the truth label $Y \in \{0, 1\}$.

**Utility.** A natural setting for a utility function, consistent with most real-world scenarios, assigns higher utility to cases where the final decision, $T$, aligns with the ground truth label, $Y$, compared to cases where $T$ and $Y$ diverge. Following Corvelo Benz & Rodriguez (2023), the utility function $u(T, Y)$ satisfies the properties as follows:

$$u(1, 1) > u(1, 0), u(1, 1) > u(0, 1), u(0, 0) > u(1, 0), u(0, 0) \geq u(0, 1). \tag{1}$$

**Cognitive disparity.** Assume there are $|\mathcal{S}|$ distinct human decision-makers groups categorized by a sensitive attribute $S \in \mathcal{S}$. To enable statistical quantification, we measure the cognition disparity (CD) between the $i$-th and $j$-th human decision-makers groups by calculating the likelihood disparities of the truth labels between different groups, despite human decision-makers having identical confidence $h$ (which reflects the different cognitive capacities to estimate the likelihood of the truth label correctly) as follows,

$$CD(i, j) = P(Y = 1 | z \in \mathcal{Z}_{h, s_i}) - P(Y = 1 | z \in \mathcal{Z}_{h, s_j}), \tag{2}$$

where $\mathcal{Z}_{h, s_i} = \{(x, h, s) | f_H(x) = h, s = s_i\}$ represents the subset of decisions made by human decision-makers in group $s_i$ who have a confidence level of $h$. Human decision-makers are referred to as cognitive-heterogeneous if $\exists i, j \in \{1, ..., |\mathcal{S}|\}, CD(i, j) \neq 0$.

**Utility disparity.** We aim to achieve equal utility across different human decision-maker groups despite disparities in their cognitive capacities. Specifically, for any given AI-provided confidence $a$ and human confidence $h$, which result in human decision-makers having identical final decision probabilities $P(T = 1)$, the utility disparity (UD) across the human-decision maker groups is expected to approach zero as follows:

$$UD = \left( \sum_{i, j \in \{1, ..., |\mathcal{S}|\}, j < i} \left| \mathbb{E}_\pi \left[ u(T, Y) | a, z \in \mathcal{Z}_{h, s_i} \right] - \mathbb{E}_\pi \left[ u(T, Y) | a, z \in \mathcal{Z}_{h, s_j} \right] \right| \right) / \binom{|\mathcal{S}|}{2} \rightarrow 0. \tag{3}$$

Specifically, in scenarios involving binary sensitive attributes, the Eq. (3) can be simplified to:

$$UD = \left| \mathbb{E}_\pi \left[ u(T, Y) | f_A(z) = a, z \in \mathcal{Z}_{h, 1} \right] - \mathbb{E}_\pi \left[ u(T, Y) | f_A(z) = a, z \in \mathcal{Z}_{h, 0} \right] \right| \rightarrow 0. \tag{4}$$

**Monotone.** When the human decision-makers act rationally, increasing human decision-makers confidence $h$ and AI confidence $a$ raises the probability of human decision-makers making positive final decisions. Regarding this assumption, we provide a detailed discussion of its rationale and scope in Appendix A.1, verify that Assumption 2.1 holds broadly in real-world AI-assisted decision-making contexts, and validate the robustness of the subsequent method under certain violations of the monotonicity assumption.

**Assumption 2.1.** (Monotone Decision Policy in AI-Assisted Decision Making) Assume that human decision-makers are rational. The decision policy is monotone, meaning that for any AI confidence $a_1$ and $a_2$, and any human decision-makers confidence $h_1$ and $h_2$, if $a_1 \leq a_2$ and $h_1 \leq h_2$, then,

$$P(T = 1 | h_1, a_1) \leq P(T = 1 | h_2, a_2). \tag{5}$$

# 3 CAN FAIR UTILITY BE ACHIEVED IN AI-ASSISTED DECISION-MAKING?

## 3.1 LIMITATION OF $\alpha_y$-CALIBRATION AND $\alpha_h$-HUMAN-ALIGNMENT

When the AI model produce confidence estimates that accurately represent the distribution of truth labels, it achieves perfect calibration Pakdaman Naeini et al. (2015); Yin et al. (2019); Zhang et al. (2020). We adopt the statistical notion of $\alpha_y$-calibration introduced by Hebert-Johnson et al. Hebert-Johnson et al. (2018), which transitions from approximate calibration to perfect calibration by adjusting the hyperparameter $\alpha_y$ from 1 to 0.

**Definition 3.1.** ($\alpha_y$-Calibration) An AI system with a confidence function $f_A : \mathcal{Z} \to [0,1]$ where $\mathcal{Z} = \{\mathcal{X}, \mathcal{H}, \mathcal{S}\}$ satisfies $\alpha_y$-calibration with respect to $\mathcal{Z}$ if there exists $\mathcal{Z}' \subseteq \mathcal{Z}$ with $|\mathcal{Z}'| \geq (1 - \alpha_y) \cdot |\mathcal{Z}|$, such that for any AI confidence $a \in [0,1]$, it holds that:

$$|P(Y = 1 \mid f_A(z) = a, z \in \mathcal{Z}') - a| \leq \alpha_y. \tag{6}$$

The Definition 3.1 bounds the proportion of samples where the difference between the AI confidence and the positive label likelihood exceeds $\alpha_y$ to be less than $\alpha_y$. When the AI confidence is perfectly calibrated ($\alpha_y \to 0$), it implies that, for the entire sample space $\mathcal{Z}$, the AI confidence $f_A$ aligns exactly with the likelihood of the positive label. Based on this definition, we demonstrate the limitation of calibration in achieving fair utility, as in Theorem 3.2.

**Theorem 3.2.** *(Utility disparity under calibration (Proof in Appendix A.3)) Given an AI-assisted decision-making with utility function $u(T, Y)$ in Eq. (1) and the human decision-makers with any monotone AI-assisted decision policy $\pi \in \Pi(H, A)$. Even when the AI confidence function $f_A$ is perfectly calibrated, if the cognitive disparity in Eq. (2) is **non-zero**, then there exists AI confidence $a$ and human confidence $h$ such that the utility disparity is also **non-zero**.*

$$\left| \mathbb{E}_\pi \left[ u(T, Y) | f_A(z) = a, z \in \mathcal{Z}_{h,1} \right] - \mathbb{E}_\pi \left[ u(T, Y) | f_A(z) = a, z \in \mathcal{Z}_{h,0} \right] \right| \neq 0. \tag{7}$$

Corvelo Benz & Rodriguez (2023) argued that $\alpha_y$-calibration fails to ensure optimal utility for monotone policies and proposed *human-alignment* as a new confidence alignment objective. They demonstrated that a perfectly human-aligned confidence function guarantees the existence of a monotone policy $\pi$ achieving optimal utility. Inspired by the superior performance of *human-alignment*, we further analyze its resulting utility disparity.

**Definition 3.3.** ($\alpha_h$-Human-alignment) An AI system with a confidence function $f_A : \mathcal{Z} \to [0,1]$ where $\mathcal{Z} = \{\mathcal{X}, \mathcal{H}, \mathcal{S}\}$, satisfies $\alpha_h$-alignment with respect to human decision-maker confidence function $f_H : \mathcal{X} \to [0,1]$ if, for any $h \in \mathcal{H}$, there exists $\mathcal{Z}'_h \subseteq \mathcal{Z}_h$ with $\mathcal{Z}_h = \{(x, h, s) \in \mathcal{Z} | f_H(x) = h\}$ and $|\mathcal{Z}'_h| \geq (1 - \alpha_h/2) \cdot |\mathcal{Z}_h|$, such that, for any $0 \leq a_1 \leq a_2 \leq 1$ and $0 \leq h_1 \leq h_2 \leq 1$, it holds that,

$$P\left(Y = 1 \mid f_A(z) = a_1, z \in \mathcal{Z}'_{h_1}\right) - P\left(Y = 1 \mid f_A(z) = a_2, z \in \mathcal{Z}'_{h_2}\right) \leq \alpha_h. \tag{8}$$

For $h_1$ and $h_2$ satisfying the monotonicity condition $h_1 \leq h_2$, the above definition bounds the violation of monotonicity in the positive label introduced by $f_A$ to at most $\alpha_h/2$ over the sample spaces $\mathcal{Z}_{h_1}$ and $\mathcal{Z}_{h_2}$. However, even if the AI confidence function $f_A$ is perfectly human-aligned ($\alpha_h \to 0$), the monotonic decision policy $\pi$ still be suboptimal in terms of fair utility.

**Theorem 3.4.** *(Utility disparity under human-alignment (Proof in Appendix A.4)) When the cognitive disparity in Eq. (2) is **non-zero**, there exist (infinitely many) AI-assisted decision-making processes with utility function $u(T, Y)$ in Eq. (1) and the human decision-maker with any monotone AI-assisted decision policy $\pi \in \Pi(H, A)$, such that even the AI confidence function $f_A$ is perfectly aligned with the human's, the AI-assisted decision-making still fails to achieve optimal utility fairness. Specifically,*

$$
\begin{aligned}
&\left| \mathbb{E}_\pi \left[ u(T, Y) | f_A(z) = a, z \in \mathcal{Z}_{h,1} \right] - \mathbb{E}_\pi \left[ u(T, Y) | f_A(z) = a, z \in \mathcal{Z}_{h,0} \right] \right| \\
&> \left| \mathbb{E}_{\pi^*} \left[ u(T, Y) | f_A(z) = a, z \in \mathcal{Z}_{h,1} \right] - \mathbb{E}_{\pi^*} \left[ u(T, Y) | f_A(z) = a, z \in \mathcal{Z}_{h,0} \right] \right|,
\end{aligned} \tag{9}
$$

*where,*

$$\pi^* = \underset{\pi \in \Pi(\mathcal{H}, \mathcal{A})}{arg\,min} \left| \mathbb{E}_\pi \left[ u(T, Y) | f_A(z) = a, z \in \mathcal{Z}_{h,1} \right] - \mathbb{E}_\pi \left[ u(T, Y) | f_A(z) = a, z \in \mathcal{Z}_{h,0} \right] \right|. \tag{10}$$

We analyze that the cause of failure in fairness arises from *human-alignment* without considering differences in the correctness of human decision-makers' confidence due to heterogeneous cognitive capacities. This discrepancy results in differing levels of *human-alignment* between groups. For groups with weaker alignment, this can lead to utility disadvantages.

It is a common occurrence in practice for there to be **non-zero cognitive disparity** among human decision-makers. Consider, for instance, a disease diagnosis scenario involving two groups of human

decision-makers: experts ($S = 1$) and general practitioners ($S = 0$) working under AI assistance. Suppose both groups diagnose patients as having the disease with a human decision-maker confidence level of $h = 0.9$. Due to differences in cognitive capacities, there may be a disparity in the true probability that the patients actually have the disease: $P(Y = 1|z \in \mathcal{Z}_{0.9,1}) > P(Y = 1|z \in \mathcal{Z}_{0.9,0})$. Based on Theorems 3.2 and 3.4, both calibrated and human-aligned AI confidence cannot guarantee utility fairness in such scenarios. In the following sections, we propose a novel alignment objective to address this issue.

## 3.2    Inter-group-alignment and utility disparity upper bound

Given the limitations of existing calibration methods in ensuring optimal fair utility, we introduce the core concept of *inter-group-alignment* in Definition 3.5. Building on this foundation, we give an upper bound on utility disparity of any AI-assisted decision making process, as in Theorem 3.6.

**Definition 3.5.** ($\alpha_g$-Inter-group-alignment) An AI system with a confidence function $f_A : \mathcal{Z} \to [0, 1]$ where $\mathcal{Z} = [\mathcal{X}, \mathcal{H}, \mathcal{S}]$, satisfies $\alpha_g$-inter-group-alignment if, for any $h \in \mathcal{H}$, there exists $\mathcal{Z}'_h \subset \mathcal{Z}_h$ with $\mathcal{Z}_h = \{(x, h, s)|f_H(x) = h\}$ and $|\mathcal{Z}'_h| \geq (1 - \alpha_g/2) \cdot |\mathcal{Z}_h|$. Let $\mathcal{Z}'_{h,s} = \{(x, h, s) \in \mathcal{Z}'_h|S = s\}$, the AI confidence is $\alpha_g$-inter-group-alignment if,

$$\left| P\left(Y = 1 \mid f_A(z) = a, z \in \mathcal{Z}'_{h,1}\right) - P\left(Y = 1 \mid f_A(z) = a, z \in \mathcal{Z}'_{h,0}\right) \right| \leq \alpha_g. \quad (11)$$

Based on Eq. (5), given identical AI confidence $a$ and human decision-maker's confidence $h$, human decision-makers will exhibit the same probability of making the final decision, $P(T = 1)$. The definition of $\alpha_g$-inter-group-alignment constrains the distribution of positive label $Y = 1$ to be statistically equal across different human decision-maker groups when $\alpha_g \to 0$. This alignment ensures that human decision-makers within each group achieve statistically similar utilities for making correct decisions.

**Theorem 3.6.** *(Utility disparity upper bound under $\alpha_h$-human-alignment and $\alpha_g$-inter-group-alignment (Proof in Appendix A.5)) For a given AI-assisted decision-making process with a utility function $u(T, Y)$ satisfying Eq. (1) and the human with any monotone AI-assisted decision policy $\pi \in \Pi(H, A)$, if the AI confidence function $f_A$ is $\alpha_h$-human-alignment and satisfies $\alpha_g$-inter-group-alignment, then the utility disparity is bounded by,*

$$\left| \mathbb{E}_\pi \left[u(T, Y)|f_A(z) = a, z \in \mathcal{Z}_{h,1}\right] - \mathbb{E}_\pi \left[u(T, Y)|f_A(z) = a, z \in \mathcal{Z}_{h,0}\right] \right|$$
$$\leq (u(1, 1) - u(0, 1) - u(1, 0) + u(0, 0)) \cdot \left(\frac{\alpha_h}{2} + \left(1 - \frac{\alpha_h}{2}\right) \cdot \left(3\alpha_g - \alpha_g^2\right)\right). \quad (12)$$

Theorem 3.6 provides a tight upper bound on the utility disparity, which is constrained by both the *human-alignment* level $\alpha_h$ and *inter-group-alignment* level $\alpha_g$. This offers valuable insights into the fairness-aware AI confidence alignment objectives, which seek to align the AI confidence as closely as possible with human decision-maker's, while simultaneously ensuring that human decision-makers from different groups, with identical confidence $h$, receive statistically similar positive label distributions when provided with the same AI confidence $a$. Based on this, we will next outline a calibration framework to achieving the dual alignment objectives simultaneously.

# 4    AI Confidence Multicalibration for Simultaneous Improvement of Utility and Fairness

**Corollary 4.1.** *For any AI-assisted decision-making with a utility function $u(T, Y)$ in Eq. (1) and is $\alpha_h$-human-alignment, the upper bound of utility disparity across different human decision-maker groups is minimized when the decision function $f_A$ satisfies perfect inter-group-alignment.*

The above corollary holds as $(3\alpha_g - \alpha_g^2) \geq 0$ for all $\alpha_g \in [0, 1]$. Consequently, under any *human-alignment* level $a_h$, the utility disparity upper bound in Theorem 3.6 is minimized when $\alpha_g = 0$. We can further refine the conditions under which the AI-assisted decision-making process provides both optimal utility and fair utility across heterogeneous human decision-maker groups as follows:

**Corollary 4.2.** *For AI-assisted decision-making processes with a utility function $u(T, Y)$ satisfying Eq. (1), if $f_A$ achieves both perfectly human-alignment and perfectly inter-group-alignment, there exist monotone AI-assisted decision policy $\pi \in \Pi(\mathcal{H}, \mathcal{A})$ that simultaneously attains optimal overall utility and fair utility among heterogeneous human decision-maker groups.*

Based on the utility disparity upper bound established in Theorem 3.6, In the following, we demonstrate how to simultaneously achieve *human-alignment* and *inter-group-alignment* through multicalibration, thereby ensuring that AI-assisted decision-making provides fair utility while guaranteeing optimal utility for all human decision-makers.

**Method 4.3.** (AI confidence $\alpha$-multicalibration accounting for cognitive disparities) Suppose human decision-makers can be partitioned into $N$ groups, with group membership values $\{s_i\}_{i=1}^N$ determined by human-sensitive attributes related to cognitive capacity. For each group $s_i$, let the collection of subsets as $\mathcal{C}_i = \{\mathcal{Z}_{h,s_i}\}_{h \in \mathcal{H}}$. An AI confidence function $f_A$ is said to satisfy $\alpha$-multicalibration accounting for cognitive disparitie if it satisfies $\alpha$-calibration (as defined in Definition 3.1) with respect to all $\mathcal{Z}_{h,s_i} \in \mathcal{C}_i$, for all $i \in \{1, \ldots, N\}$.

We refer to Method 4.3 as **cognition-aware multicalibration**, to distinguish it from the multicalibration that does not account for cognitive disparities Corvelo Benz & Rodriguez (2023), defined as follows:

**Method 4.4.** (AI confidence $\alpha$-multicalibration (ignoring cognitive disparities)) Let $\mathcal{C} = \{\mathcal{Z}_h\}_{h \in \mathcal{H}}$ be a collection of subsets over the domain $\mathcal{Z}$. An AI confidence function $f_A : \mathcal{Z} \to [0, 1]$ is said to satisfy $\alpha$-multicalibration if it satisfies $\alpha$-calibration (as defined in Definition 3.1) with respect to all $\mathcal{Z}_h \in \mathcal{C}$.

**Theorem 4.5.** *Suppose $f_A(z)$ is $\alpha/2$-cognition-aware multicalibrated with respect to the collection $\mathcal{C} = \{\mathcal{Z}_{h,s_i}\}_{h \in \mathcal{H}, s_i \in \mathcal{S}}$. Then, $f_A$ satisfy both $\alpha$-human-alignment and $\alpha$-inter-group-alignment (see proof in Appendix A.6).*

To reduce the computational burden of adjusting confidence over continuous intervals, we implement Method 4.3 using $\lambda$-discretization Hebert-Johnson et al. (2018); the full algorithm is presented in Algorithm 1. Specifically, $\lambda$-discretization partitions the confidence interval $[0, 1]$ of $f_A$ into $\lfloor 1/\lambda \rfloor$ bins of width $\lambda$, with centers at $\Lambda = \{\frac{\lambda}{2}, \frac{3\lambda}{2}, \ldots, 1 - \frac{\lambda}{2}\}$. Confidence adjustment is then applied to the sample set within each bin. According to Hebert-Johnson et al. (2018), the resulting discretized confidence function can achieve $(\widetilde{\alpha} + \lambda)$-multicalibration. In light of Theorem 4.5, to ensure $f_A$ satisfies at least $\alpha$-human-alignment and $\alpha$-inter-group-alignment, it suffices to ensure $\widetilde{\alpha} + \lambda \leq \alpha/2$.

## 5 EXPERIMENTS

### 5.1 SETTINGS

**Dataset.** We utilize a publicly available dataset for human-AI interactions across 4 tasks Vodrahalli et al. (2022a). In each task, human decision-makers first provide their confidence (used to construct $f_H$). After receiving AI advice (used to construct $f_A$), participants update their final decision confidence (used to construct $\pi(h, a)$). The tasks span different data modalities (visual, text, and tabular). ① In the Art (Image) task, participants determine the art period of a painting from two options. ② In the Cities (Image) task, participants are asked to determine the originating city of an image from a binary choice. ③ In the Sarcasm (Text) task, participants determine if a Reddit text snippet contains sarcasm. ④ In the Census (Tabular) task, participants assess whether an individual earns at least $50,000 annually based on their demographic information. In our experiments, human decision-makers are categorized into two groups by their "education" levels: Group $S = 0$, which includes individuals with at least a Master's degree, and Group $S = 1$, consists of those with a degree lower than a Master's. The data were preprocessed to exclude samples with missing information and confounding factors (see Appendix A.7.1), resulting in a final dataset of $14,999$ AI-assisted decision-making records from $469$ participants overall.

**Baselines.** ① **No Adjust**, where the original confidence values inherent in the dataset are retained without any modification; ② **Cognition-unaware Multicalibration**, where the AI confidence is adjusted using multicalibration *ignoring cognitive disparities* (see Method 4.4).

---

**Algorithm 1:** Cognition-Aware AI Confidence Multicalibration

---

**Input:** Calibration tolerance $\widetilde{\alpha}$, AI confidence bin width $\lambda$
**Output:** Calibrated AI confidence function $f_A$
Define the attributes $S_1, \ldots, S_n$ related to cognitive capacity to group human decision-makers; let
$\quad \mathbf{s} = (s_1, \ldots, s_n) \in S_1 \times \cdots \times S_n$ denote group membership
Construct $\mathcal{C} = \{\mathcal{Z}_{h,\mathbf{s}} = \{z = (x, h, s) \mid f_H(x) = h, \ s = \mathbf{s}\}\}$ for all $h \in \mathcal{H}, \mathbf{s} \in S_1 \times \cdots \times S_n$
**repeat**
    **foreach** $\mathcal{Z}_{h,\mathbf{s}} \in \mathcal{C}$ **do**
        **for** $a = 1$ **to** $\lfloor 1/\lambda \rfloor$ **do**
            Define $\mathcal{Z}_{h,\mathbf{s}}^a \leftarrow \{z \in \mathcal{Z}_{h,\mathbf{s}} \mid f_A(z) \in [\Lambda[a] - \lambda/2, \ \Lambda[a] + \lambda/2)\}$
            **if** $P(z \in \mathcal{Z}_{h,\mathbf{s}}^a) < \widetilde{\alpha}\lambda \cdot P(z \in \mathcal{Z}_{h,\mathbf{s}})$ **then**
                  $\lfloor$ **continue**
            **if** $\left| \mathbb{E}[f_A(z) \mid z \in \mathcal{Z}_{h,\mathbf{s}}^a] - P(Y = 1 \mid z \in \mathcal{Z}_{h,\mathbf{s}}^a) \right| > \widetilde{\alpha}$ **then**
                  $\lfloor$ Update $f_A(z) \leftarrow f_A(z) + \left( P(Y = 1 \mid \mathcal{Z}_{h,\mathbf{s}}^a) - \mathbb{E}[f_A(z) \mid \mathcal{Z}_{h,\mathbf{s}}^a] \right)$ for all $z \in \mathcal{Z}_{h,\mathbf{s}}^a$

**until** *no* $\mathcal{Z}_{h,\mathbf{s}}^a$ *updated*;
**for** $a = 1$ **to** $\lfloor 1/\lambda \rfloor$ **do**
    **foreach** $\mathbf{s} \in S_1 \times \cdots \times S_n$ **do**
        Let $\mathcal{Z}_{\mathbf{s}}^a = \{z = (x, h, s) \mid s = \mathbf{s}, \ f_A(z) \in [\Lambda[a] - \lambda/2, \ \Lambda[a] + \lambda/2)\}$
        Set $f_A(z) \leftarrow \mathbb{E}[f_A(z) \mid z \in \mathcal{Z}_{\mathbf{s}}^a]$ for all $z \in \mathcal{Z}_{\mathbf{s}}^a$

**return** $f_A$

---

**Hyperparameters.** We configure the hyperparameters as follows: $\widetilde{\alpha} = 0.0001$ and $\lambda = 0.125$, ensuring that the level of cognition-aware multicalibration is approximately $0.125$. We report the impact of hyperparameters $\widetilde{\alpha}$ and $\lambda$ on alignment in Appendix A.8.3, showing that smaller $\widetilde{\alpha} + \lambda$ values lead to improved human-alignment and inter-group-alignment.

**Decision policy function.** Since the dataset only provides uncalibrated confidences, we evaluate AI-assisted decision performance after calibration by learning the decision policy $\pi(h, a)$ using a multi-layer perceptron (MLP) classifier with one hidden layer of 20 nodes and ReLU activation.

**Experimental setup.** ① **Alignment quantification**: We evaluate how effectively *Cognition-aware Multicalibration* achieves both *human-alignment* and *inter-group-alignment*. ② **AI-assisted decision utility**: We assess whether *Cognition-aware Multicalibration* improves both expected utility and utility fairness of the final decision policy. ③ **Ablation study**: We compare the expected utility and disparity of human-only decisions $\pi(h)$ and AI-only decisions $\pi(a)$ to understand how *Cognition-aware Multicalibration* contributes to fairness across diverse decision-maker groups. ④**Additional experiment**: Due to page limitations, we provide additional experimental results in Appendix A.8, including quantitative ablation study results (Appendix A.8.1), group-level accuracy distributions under fairness considerations (Appendix A.8.2), the impact of discretization parameters on calibration performance (Appendix A.8.3), generalization to multiple groups (Appendix A.8.4), computational complexity analysis (Appendix A.8.5), and multi-class classification scenarios (Appendix A.8.6), further validating the effectiveness and robustness of the proposed method.

**Evaluation metric.** We use accuracy to evaluate the decision utility and, naturally, evaluate the fair utility to diverse human decision-maker groups by measuring accuracy disparities as follows,

$$\text{Disp} = \mathbb{E}\left[\mathbf{1}(T = Y) \mid s = 1\right] - \mathbb{E}\left[\mathbf{1}(T = Y) \mid s = 0\right]. \tag{13}$$

In terms of measuring *human-alignment*, Corvelo Benz & Rodriguez (2023) proposed two discrete metrics: the expected alignment error (EAE) and the maximum alignment error (MAE). The discretization details are in the Appendix A.7.2. The limitation of metrics based on discretization lies in the finite number of discrete intervals, which may fail to capture alignment across the entire continuous confidence space accurately. However, significant variations in the metrics can reflect variations in alignment more accurately. In such cases, the influence of unmeasured alignment on the overall results is reduced.

$$\text{EAE} = \max\left(0, \ \frac{1}{N} \cdot \sum_{i \leq i', j \leq j'} \left[ P\left(Y = 1 \mid z \in \mathcal{Z}_i^j\right) - P\left(Y = 1 \mid z \in \mathcal{Z}_{i'}^{j'}\right) \right]\right). \tag{14}$$

$$\text{MAE} = \max\left(0, \ \max_{i \leq i', j \leq j'} \left( P\left(Y = 1 \mid z \in \mathcal{Z}_i^j\right) - P\left(Y = 1 \mid z \in \mathcal{Z}_{i'}^{j'}\right) \right)\right). \tag{15}$$

Table 1: Alignment quantification results (**Bold**: best result; Black: mean; Blue: standard deviation). Lower values indicate better alignment for all metrics.

| Exp. | No Adjust | | | | Method 4.4 | | | | Method 4.3 | | | |
|---|---|---|---|---|---|---|---|---|---|---|---|---|
| | EAE | MAE | EIAE | MIAE | EAE | MAE | EIAE | MIAE | EAE | MAE | EIAE | MIAE |
| 1 | 0.0010 | 0.1167 | 0.0525 | 0.4286 | **0.0009** | **0.0494** | 0.0345 | 0.2291 | 0.0025 | 0.0675 | **0.0209** | **0.1360** |
| | (6.5E-19) | (1.5E-16) | (1.2E-16) | (5.0E-16) | (9.7E-19) | (6.9E-18) | (2.1E-17) | (5.5E-17) | (1.7E-18) | (9.7E-17) | (2.7E-17) | (2.5E-16) |
| 2 | 0.0047 | 0.2239 | 0.1094 | 0.4085 | **0.0003** | **0.0179** | 0.0145 | 0.2609 | 0.0007 | 0.0493 | **0.0031** | **0.1132** |
| | (6.9E-18) | (3.3E-16) | (1.9E-16) | (6.6E-16) | (7.5E-19) | (2.5E-17) | (2.6E-17) | (3.3E-16) | (1.1E-19) | (2.7E-17) | (3.9E-18) | (2.3E-16) |
| 3 | 0.0008 | 0.0830 | 0.1180 | 0.5702 | **0.0006** | **0.0576** | 0.0784 | 0.5702 | 0.0013 | 0.0989 | **0.0063** | **0.0458** |
| | (4.3E-19) | (1.2E-16) | (1.9E-16) | (1.1E-16) | (1.5E-18) | (6.2E-17) | (6.9E-17) | (1.1E-16) | (1.5E-18) | (2.4E-16) | (1.6E-17) | (7.6E-17) |
| 4 | 0.0091 | 0.2985 | 0.0794 | 0.3601 | **0.0002** | **0.0195** | 0.0328 | 0.5275 | 0.0005 | 0.0410 | **0.0072** | **0.1062** |
| | (1.7E-18) | (4.4E-16) | (1.2E-16) | (1.1E-16) | (1.1E-19) | (2.7E-17) | (3.4E-17) | (1.3E-15) | (3.2E-19) | (4.2E-17) | (7.8E-18) | (6.9E-17) |

Table 2: T-tests on the performance of *cognition-unaware multicalibration* and *cognition-aware multicalibration* under 100 repeated experiments.

| Exp. | Utility | | Utility Disparity | |
|---|---|---|---|---|
| | t | p-value | t | p-value |
| 1 | 3.018 | 0.003 | 9.486 | 0.000 |
| 2 | 0.345 | 0.731 | 15.484 | 0.000 |
| 3 | -12.187 | 0.000 | 3.186 | 0.002 |
| 4 | 4.839 | 0.000 | 27.556 | 0.000 |

We develop two metrics for measuring *inter-group-alignment*: the expected *inter-group-alignment* error (EIAE) and the maximum *inter-group-alignment* error (MIAE).

$$\text{EIAE} = \frac{1}{N} \cdot \sum_{i,j} \left| P\left(Y = 1 \mid z \in \mathcal{Z}_{i,1}^j\right) - P\left(Y = 1 \mid z \in \mathcal{Z}_{i,0}^j\right) \right|, \tag{16}$$

$$\text{MIAE} = \max_{i,j} \left\{ \left| P\left(Y = 1 \mid z \in \mathcal{Z}_{i,1}^j\right) - P\left(Y = 1 \mid z \in \mathcal{Z}_{i,0}^j\right) \right| \right\}, \tag{17}$$

where $Z_{i,0}^j$ and $Z_{i,1}^j$ contain samples from the groups $S = 0$ and $S = 1$, respectively, located in the $(i, j)$-th cell of the grid formed by the discretization of human confidence and AI confidence.

## 5.2 RESULTS

The key takeaway from the experimental results presented in Table 1 is **the proposed *Cognition-aware Multicalibration*'s effectiveness in maintaining competitive *human-alignment* while significantly improving *inter-group-alignment***. In the Cities and Centus tasks, both Method 4.4 and Method 4.3 significantly reduce EAE and MAE compared to the *No Adjust* case, indicating better alignment with human decision-maker's confidence. In the Art and Sarcasm tasks, while *Cognition-aware Multicalibration* exhibits slightly higher EAE and MAE values, the differences remain marginal with magnitudes of $10^{-3}$ and $10^{-2}$, respectively. More importantly, when evaluating *inter-group-alignment* using EIAE and MIAE metrics, *Cognition-aware Multicalibration* consistently achieves the best performance across all four tasks, substantially reducing both EIAE and MIAE compared to both the *No Adjust* baseline and *Cognition-unaware Multicalibration*.

Figure 1 and Table 2 yield three key takeaways: ① *Cognition-unaware Multicalibration* **fails to improve fairness and sometimes worsens it**, as in task 1 where utility disparity exceeds the *No Adjust* baseline; ② **Both calibration methods consistently improve decision utility over *No Adjust* across all tasks**, with T-tests showing comparable performance between them; ③ *Cognition-aware Multicalibration* **consistently achieves the lowest utility disparity across all tasks**, substantially reducing group disparities compared to both baselines. Using an MLP-based decision model, we report the distribution of final decisions over 100 trials with random seeds ($0 - 99$). The *No Adjust* case shows no variance, as its decisions are fixed by the dataset.

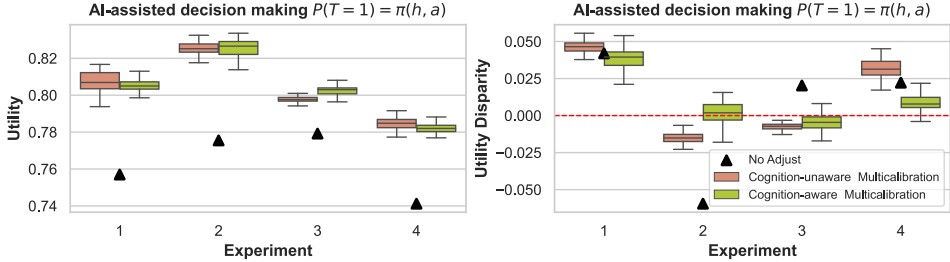

Figure 1: Statistics of utility and utility disparity over 100 experiments, where the final decision $P(T = 1) = \pi(h, a)$ is made by human with AI assistance. The AI confidence is adjusted by *No Adjust*, *Cognition-unaware Multicalibration* and *Cognition-aware Multicalibration*, respectively. Higher utility and lower utility disparity indicate better performance.

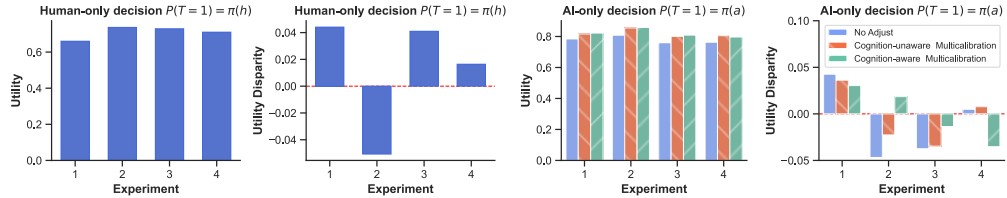

Figure 2: The utility and utility disparity where the final decision is made by human-only $P(T = 1) = \pi(h)$ or AI-only $P(T = 1) = \pi(a)$.

In Figure 2, we provide the utility and utility disparity when final decisions are made solely by human decision-makers and AI independently. The key observation is that ***Cognition-aware Multicalibration* adjusts the AI's confidence to mitigate utility disparity caused by human-only decisions, either by reducing the disparity or by offsetting disparity with the opposite sign**. This capability is absent in *Cognition-unaware Multicalibration*, which in some cases (e.g., the Sarcasm task) can even worsen utility disparity compared to human-only decisions.

## 6 RELATED WORK

AI confidence helps decision-makers calibrate their trust in the AI and appropriately apply AI knowledge to make final decisions, especially in cases where the AI model is likely to perform poorly. To enhance human decision-maker's comprehension of AI prediction uncertainty, AI model confidence is primarily calibrated to reflect the probabilities of classification correctness Hebert-Johnson et al. (2018); Guo et al. (2017); Zhao et al. (2021). However, experimental evidence by Vodrahalli et al. (2022b) indicated that AI models, when perceived as more confident than they actually are—rather than being well-calibrated—can enhance the accuracy of final decisions made by human decision-makers after considering AI advice. The work most closely related to ours is Corvelo Benz & Rodriguez (2023), which conducted a systematic theoretical analysis of scenarios where well-calibrated AI confidence may lead to suboptimal utility for rational decision-makers. They also introduced the concept of AI confidence *human-alignment*, enabling rational decision-makers to achieve optimal utility. Previous works in AI-assisted decision making assume that human decision-makers are homogeneous, overlooking heterogeneity Rambachan (2024); Rambachan et al. (2022); De-Arteaga et al. (2021) in their cognitive capacities. When AI assistance is used as a new information resource, failing to account for cognitive heterogeneity can exacerbate utility disparities among decision-makers, exacerbating societal inequities. Distinguished from Corvelo Benz & Rodriguez (2023), our work addresses a novel and previously unexplored dimension of AI-assisted decision-making: ensuring equitable utility for human decision-makers with varying levels of cognitive ability. Furthermore, we contribute to a solid theoretical framework for analyzing and mitigating fairness issues in AI-assisted decision-making. Different from the commonly used algorithmic fairness in the fair machine learning area, which mainly aims to ensure unbiased outcomes for individuals being decided upon (e.g., patients), our focus is on ensuring fairness for the decision-makers (e.g.,

doctors). Pleiss et al. (2017) investigated the compatibility between calibration and equalized odds. Beyond differences in application contexts—where their focus is on predictor scenarios rather than AI-assisted decision-making—our work also targets a distinct fairness concept. This fairness concept shares some similarities with accuracy disparity in centralized model training Chi et al. (2021) and egalitarian fairness in decentralized learning Donahue & Kleinberg (2023), but it requires fundamentally different solving methods tailored to the context of AI-assisted decision-making.

## 7 Conclusions and Limitations

In this work, we systematically analyzed the issue of unfair utility that arises when human decision-makers with heterogeneous cognitive capacities engage in AI-assisted decision-making. We identified that rational decision-makers may not achieve equal utility under existing AI confidence calibration methods. To address this issue, we introduced a novel AI confidence adjustment objective, *inter-group-alignment*, which, when combined with *human-alignment*, establishes an upper bound on utility disparities. Building on this foundation, we proposed *cognition-aware multicalibration*, which ensures interpretable utility fairness and optimal overall utility, as it serves as a sufficient condition for simultaneously achieving both *human alignment* and *inter-group-alignment*. Experiments conducted on real datasets thoroughly evaluated the effectiveness of our proposed AI confidence adjustment approach in providing optimal and fair utility across human decision-makers with heterogeneous cognitive capacities. Our study offered a new perspective on fairness concerns in AI-assisted decision-making. As the first work in this area, we acknowledged that future research can explore additional fairness metrics, live user studies, and dynamic human behavioral evolution in long-term interactions. Due to space limitations, these aspects are discussed in Appendix A.9.

### Acknowledgments

This work was supported in part by Major Program of Guangdong Province under Grant 2021QN02X166, and in part by the National Natural Science Foundation of China (Project No. 72031003). Any opinions, findings, and conclusions or recommendations expressed in this material are those of the author(s) and do not necessarily reflect the views of the funding parties.

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

## A    Technical Appendices

### A.1    Rationale and Scope of Assumption 2.1

Assumption 2.1 describes the behavior of rational decision-makers who aim to maximize decision utility: they increase their likelihood of making a positive decision $T = 1$ when both their own confidence $h$ and AI confidence $a$ increase. While Assumption 2.1 could be violated in atypical scenarios—such as AI aversion, where decision-makers systematically decrease their decision likelihood as AI confidence increases, or random errors where decision-makers accidentally make incorrect decisions—such cases represent departures from rational utility-maximizing behavior and are outside our current scope.

**Assumption 2.1 violation rate evaluation.** To empirically assess the prevalence of rational behavior in real-world settings, we evaluated Assumption 2.1 across all four real-world AI-assisted decision-making tasks by computing pairwise violation rates. Specifically, for all pairs of instances $(i, j)$ where $h_i \leq h_j$ and $a_i \leq a_j$, we checked whether $P(T_i = 1) > P(T_j = 1)$ (a monotonicity violation). Table 3 reports the results. The violation rates are consistently low across all datasets. This provides strong empirical support that Assumption 2.1 holds generally in real-world AI-assisted decision-making contexts.

**Performance under increased violation rate.** To validate robustness under monotonicity violations, we sampled a instances set with increased Assumption 2.1 violating rates. The violation rates

Table 3: Assumption 2.1 violation rates across $4$ tasks.

| Exp. | Art | Cities | Sarcasm | Census |
|---|---|---|---|---|
| Violation Rate | 0.53% | 0.41% | 0.52% | 0.20% |

Table 4: Assumption 2.1 violation rates across $4$ tasks in sampled set.

| Exp. | Art | Cities | Sarcasm | Census |
|---|---|---|---|---|
| Violation Rate | 4.7% | 2.7% | 1.9% | 4.9% |

after sampling are shown in Table 4. Performance comparisons on this "noisy" dataset are presented in Figure 3 and Table 5. Results show that *Cognition-aware Multicalibration* maintains competitive utility compared to *Cognition-unaware Multicalibration* while significantly improving fairness by reducing utility disparities.

## A.2 PRE-LEMMAS

**Lemma A.1.** *If the utility function $u$ satisfies Eq. (1) and the distribution of $Y = 1$ satisfies $P(Y = 1|S = 1) > P(Y = 1|S = 0)$, then a trivial policy $\pi$ that always decides $T = 1$ will consistently result in a positive utility disparity, while a trivial policy that always decides $T = 0$ will consistently result in a negative utility disparity.*

$$\mathbb{E}_{Y \in P}\left[u(1, Y)|S = 1\right] - \mathbb{E}_{Y \in P}\left[u(1, Y)|S = 0\right] > 0, \tag{18}$$

$$\mathbb{E}_{Y \in P}\left[u(0, Y)|S = 1\right] - \mathbb{E}_{Y \in P}\left[u(0, Y)|S = 0\right] \leq 0. \tag{19}$$

**Lemma A.2.** *If the utility function $u$ satisfies Eq. (1) and the distribution of $Y = 1$ satisfies $P(Y = 1|S = 1) < P(Y = 1|S = 0)$, then a trivial policy $\pi$ that always decides $T = 1$ will consistently result in a negative utility disparity, while a trivial policy that always decides $T = 0$ will consistently result in a positive utility disparity.*

$$\mathbb{E}_{Y \in P}\left[u(1, Y)|S = 1\right] - \mathbb{E}_{Y \in P}\left[u(1, Y)|S = 0\right] < 0, \tag{20}$$

$$\mathbb{E}_{Y \in P}\left[u(0, Y)|S = 1\right] - \mathbb{E}_{Y \in P}\left[u(0, Y)|S = 0\right] \geq 0. \tag{21}$$

***Proof.*** *For Lemma A.1: As $u(1, 1) > u(1, 0)$ and $u(0, 0) \geq u(0, 1)$, when $P(Y = 1|S = 1) > P(Y = 1|S = 0)$, we have*

$$\begin{aligned}
&\mathbb{E}_{Y \in P}\left[u(1, Y)|S = 1\right] - \mathbb{E}_{Y \in P}\left[u(1, Y)|S = 0\right] \\
&= P(Y = 1|S = 1) \cdot u(1, 1) + (1 - P(Y = 1|S = 1)) \cdot u(1, 0) \\
&- P(Y = 1|S = 0) \cdot u(1, 1) - (1 - P(Y = 1|S = 0)) \cdot u(1, 0) \\
&= (P(Y = 1|S = 1) - P(Y = 1|S = 0)) \cdot (u(1, 1) - u(1, 0)) > 0.
\end{aligned} \tag{22}$$

$$\begin{aligned}
&\mathbb{E}_{Y \in P}\left[u(0, Y)|S = 1\right] - \mathbb{E}_{Y \in P}\left[u(0, Y)|S = 0\right] \\
&= P(Y = 1|S = 1) \cdot u(0, 1) + (1 - P(Y = 1|S = 1)) \cdot u(0, 0) \\
&- P(Y = 1|S = 0) \cdot u(0, 1) - (1 - P(Y = 1|S = 0)) \cdot u(0, 0) \\
&= (P(Y = 1|S = 1) - P(Y = 1|S = 0)) \cdot (u(0, 1) - u(0, 0)) \leq 0.
\end{aligned} \tag{23}$$

*For Lemma A.2: Similarly, when $P(Y = 1|S = 1) < P(Y = 1|S = 0)$, we have,*

$$\begin{aligned}
&\mathbb{E}_{Y \in P}\left[u(1, Y)|S = 1\right] - \mathbb{E}_{Y \in P}\left[u(1, Y)|S = 0\right] \\
&= (P(Y = 1|S = 1) - P(Y = 1|S = 0)) \cdot (u(1, 1) - u(1, 0)) < 0.
\end{aligned} \tag{24}$$

$$\begin{aligned}
&\mathbb{E}_{Y \in P}\left[u(0, Y)|S = 1\right] - \mathbb{E}_{Y \in P}\left[u(0, Y)|S = 0\right] \\
&= (P(Y = 1|S = 1) - P(Y = 1|S = 0)) \cdot (u(0, 1) - u(0, 0)) \geq 0.
\end{aligned} \tag{25}$$

$\square$

Table 5: T-tests on the performance of *cognition-unaware multicalibration* and *cognition-aware multicalibration* under $100$ repeated experiments.

| Exp. | Utility | | Utility Disparity | |
|---|---|---|---|---|
| | t | p-value | t | p-value |
| 1 | -5.350 | 0.000 | 16.789 | 0.000 |
| 2 | -3.571 | 0.002 | 6.949 | 0.000 |
| 3 | -19.342 | 0.000 | 55.487 | 0.000 |
| 4 | 1.782 | 0.092 | 1.010 | 0.326 |

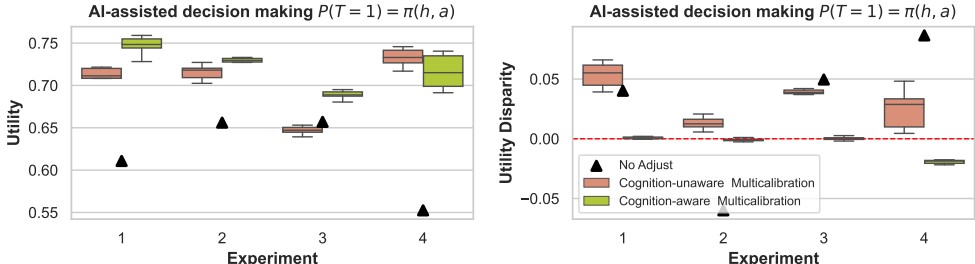

Figure 3: Performance with increased monotonicity violation rates. Grouped by "education": Statistics of utility and utility disparity over 100 experiments, where the final decision $P(T = 1) = \pi(h, a)$ is made by human with AI assistance. The AI confidence is adjusted by *No Adjust*, *Cognition-unaware Multicalibration*, and *Cognition-aware Multicalibration*, respectively.

### A.3 PROOF OF THEOREM 3.2.

**Theorem 3.2** (Utility disparity under calibration) Given an AI-assisted decision-making with utility function $u(T, Y)$ in Eq. (1) and the human decision-makers with any monotone AI-assisted decision policy $\pi \in \Pi(H, A)$. Even when the AI confidence function $f_A$ is perfectly calibrated, if the cognitive disparity in Eq. (2) is **non-zero**, then there exists AI confidence $a$ and human confidence $h$ such that the utility disparity is also **non-zero**.

$$
\Big| \mathbb{E}_\pi \left[ u(T, Y) | f_A(z) = a, z \in \mathcal{Z}_{h,1} \right]
$$
$$
- \mathbb{E}_\pi \left[ u(T, Y) | f_A(z) = a, z \in \mathcal{Z}_{h,0} \right] \Big| \neq 0. \tag{26}
$$

***Proof.*** *According to the law of total expectation, and the final decision $P(T = 1) = \pi(h, a)$ independent of the sensitive attribute $S$ (consistent with reality that different human decision-makers make decisions based solely on their own confidence and those of the AI Corvelo Benz & Rodriguez (2023)), the expected utility disparity can be formulated as follows:*

$$
\left| \mathbb{E}_\pi \left[ u(T,Y) | f_A(z) = a, z \in \mathcal{Z}_{h,1} \right] - \mathbb{E}_\pi \left[ u(T,Y) | f_A(z) = a, z \in \mathcal{Z}_{h,0} \right] \right|
$$
$$
= \left| P(Y = 1 | f_A(z) = a, z \in \mathcal{Z}_{h,1}) - P(Y = 1 | f_A(z) = a, z \in \mathcal{Z}_{h,0}) \right|
$$
$$
\cdot \left| (u(1,1) - u(1,0) - u(0,1) + u(0,0)) \cdot P_\pi(T = 1 | f_A(z) = a, z \in \mathcal{Z}_h) - (u(0,0) - u(0,1)) \right|. \tag{27}
$$

*To prove the above equation, let's first look at $\mathbb{E}_\pi \left[ u(T,Y) | f_A(z) = a, z \in \mathcal{Z}_{h,1} \right]$.*

$$
\mathbb{E}_\pi \left[ u(T,Y) | f_A(z) = a, z \in \mathcal{Z}_{h,1} \right]
$$
$$
= \mathbb{E} \left[ u(1,Y) | f_A(z) = a, z \in \mathcal{Z}_{h,1} \right] \cdot P_\pi(T = 1 | f_A(z) = a, z \in \mathcal{Z}_h)
$$
$$
+ \mathbb{E} \left[ u(0,Y) | f_A(z) = a, z \in \mathcal{Z}_{h,1} \right] \cdot \left[ 1 - P_\pi(T = 1 | f_A(z) = a, z \in \mathcal{Z}_h) \right]
$$
$$
= P_\pi(T = 1 | f_A(z) = a, z \in \mathcal{Z}_h)
$$
$$
\cdot \left[ \mathbb{E} \left[ u(1,Y) | f_A(z) = a, z \in \mathcal{Z}_{h,1} \right] - \mathbb{E} \left[ u(0,Y) | f_A(z) = a, z \in \mathcal{Z}_{h,1} \right] \right]
$$
$$
+ \mathbb{E} \left[ u(0,Y) | f_A(z) = a, z \in \mathcal{Z}_{h,1} \right]
$$

*Similarly, $\mathbb{E}_\pi [u(T, Y)|f_A(z) = a, z \in \mathcal{Z}_{h,0}]$ can be formulated as followed.*

$$\mathbb{E}_\pi [u(T, Y)|f_A(z) = a, z \in \mathcal{Z}_{h,0}]$$
$$= P_\pi(T = 1|f_A(z) = a, z \in \mathcal{Z}_h)$$
$$\cdot \left[ \mathbb{E} [u(1, Y)|f_A(z) = a, z \in \mathcal{Z}_{h,0}] - \mathbb{E} [u(0, Y)|f_A(z) = a, z \in \mathcal{Z}_{h,0}] \right]$$
$$+ \mathbb{E} [u(0, Y)|f_A(z) = a, z \in \mathcal{Z}_{h,0}]$$

*Therefore, this equation $|\mathbb{E}_\pi [u(T, Y)|f_A(z) = a, z \in \mathcal{Z}_{h,1}] - \mathbb{E}_\pi [u(T, Y)|f_A(z) = a, z \in \mathcal{Z}_{h,0}]|$ can be expanded into the following form.*

$$|\mathbb{E}_\pi [u(T, Y)|f_A(z) = a, z \in \mathcal{Z}_{h,1}] - \mathbb{E}_\pi [u(T, Y)|f_A(z) = a, z \in \mathcal{Z}_{h,0}]|$$
$$= |P(Y = 1|f_A(z) = a, z \in \mathcal{Z}_{h,1}) - P(Y = 1|f_A(z) = a, z \in \mathcal{Z}_{h,0})|$$
$$\cdot |[u(1, 1) - u(1, 0) - u(0, 1) + u(0, 0)] \cdot P_\pi(T = 1|f_A(z) = a, z \in \mathcal{Z}_h) - [u(0, 0) - u(0, 1)]| \,.$$

*Based on the law of total probability, if the cognitive disparity in Eq. (2) is **non-zero**, there exists an AI confidence $a$ such that,*

$$P(Y = 1|f_A(z) = a, z \in \mathcal{Z}_{h,1}) - P(Y = 1|f_A(z) = a, z \in \mathcal{Z}_{h,0}) \neq 0. \tag{28}$$

*Let*

$$\mathbf{Q} = |[u(1, 1) - u(1, 0) - u(0, 1) + u(0, 0)] \cdot P_\pi(T = 1|f_A(z) = a, z \in \mathcal{Z}_h) - [u(0, 0) - u(0, 1)]|$$
$$\sim [0, \max(u(0, 0) - u(0, 1), u(1, 1) - u(1, 0))] \,.$$

*Given that the decision policy $\pi(h, a)$ is monotone, it holds that, for any given AI confidence $a$,*

$$\exists h : P_\pi(T = 1|f_A(z) = a, z \in \mathcal{Z}_h) \neq \frac{u(0, 0) - u(0, 1)}{u(1, 1) - u(1, 0) - u(0, 1) + u(0, 0)} \rightarrow \mathbf{Q} \neq 0. \tag{29}$$

*Therefore, there exists AI confidence $a$ and human confidence $h$ such that,*

$$|\mathbb{E}_\pi [u(T, Y)|f_A(z) = a, z \in \mathcal{Z}_{h,1}] - \mathbb{E}_\pi [u(T, Y)|f_A(z) = a, z \in \mathcal{Z}_{h,0}]| \neq 0.$$

*Thus, Theorem 3.2 is proven.* □

## A.4   PROOF OF THEOREM 3.4

**Theorem 3.4** (Utility disparity under human-alignment) When the cognitive disparity in Eq. (2) is **non-zero**, there exist (infinitely many) AI-assisted decision-making processes with utility function $u(T, Y)$ in Eq. (1) and the human decision-maker with any monotone AI-assisted decision policy $\pi \in \Pi(H, A)$, such that even the AI confidence function $f_A$ is perfectly aligned with the human's, the AI-assisted decision-making still fails to achieve optimal utility fairness. Specifically,

$$|\mathbb{E}_\pi [u(T, Y)|f_A(z) = a, z \in \mathcal{Z}_{h,1}] - \mathbb{E}_\pi [u(T, Y)|f_A(z) = a, z \in \mathcal{Z}_{h,0}]|$$
$$> |\mathbb{E}_{\pi^*} [u(T, Y)|f_A(z) = a, z \in \mathcal{Z}_{h,1}] - \mathbb{E}_{\pi^*} [u(T, Y)|f_A(z) = a, z \in \mathcal{Z}_{h,0}]| \,. \tag{30}$$

where,

$$\pi^* = \underset{\pi \in \Pi(\mathcal{H}, \mathcal{A})}{\arg\min} |\mathbb{E}_\pi [u(T, Y)|f_A(z) = a, z \in \mathcal{Z}_{h,1}] - \mathbb{E}_\pi [u(T, Y)|f_A(z) = a, z \in \mathcal{Z}_{h,0}]| \,. \tag{31}$$

**Proof.**   *We first define $\bar{a}$, which represents the smallest AI system's confidence value for given confidence level $h$, such that,*

$$\bar{a} = \min \left\{ a \in \mathcal{A} \mid P(Y = 1 \mid f_A(z) = a, z \in \mathcal{Z}_{h,1}) - P(Y = 1 \mid f_A(z) = a, z \in \mathcal{Z}_{h,0}) > 0 \right\}. \tag{32}$$

*We demonstrate through the following four cases that there are infinitely many AI-assisted decision-making processes where, despite the AI confidence being human-aligned, the AI-assisted system fails to achieve optimal utility disparity.*

**Case 1.** *For any confidence $[h_1, a_1]$ with $a_1 < \bar{a}_1$, according to Eq. (32), it holds that,*

$$P(Y = 1 \mid f_A(z) = a_1, z \in \mathcal{Z}_{h_1,1}) - P(Y = 1 \mid f_A(z) = a_1, z \in \mathcal{Z}_{h_1,0}) \leq 0. \tag{33}$$

*Furthermore, there exists another $[h_2, a_2]$, where $a_2 > \max(\bar{a}_2, a_1)$ and $h_2 > h_1$ such that,*

$$P(Y = 1 \mid f_A(z) = a_2, z \in \mathcal{Z}_{h_2,1}) - P(Y = 1 \mid f_A(z) = a_2, z \in \mathcal{Z}_{h_2,0}) > 0. \tag{34}$$

*In the case where $f_A$ is $\alpha_h$-alignment with respect to $f_H$, according to Definition 3.3, for any $h \in \mathcal{H}$, there exists $\mathcal{Z}_h' \subset \mathcal{Z}_h$ with $\mathcal{Z}_h = \{(x, h, s) | f_H(x) = h\} \subset \mathcal{Z}$ and $|\mathcal{Z}_h'| \geq (1 - \alpha_h/2) \cdot |\mathcal{Z}_h|$ such that,*

$$P(Y = 1 | a_1, z \in \mathcal{Z}_{h_1}') - P(Y = 1 | a_2, z \in \mathcal{Z}_{h_2}') = \alpha^* < \alpha, \alpha = \max(0, \alpha^*). \tag{35}$$

*Based on the law of total probability, the Eq. (35) can be expanded as follows:*

$$
\begin{aligned}
&P(Y = 1 | a_1, z \in \mathcal{Z}_{h_1,1}') \cdot P(S = 1 | a_1, z \in \mathcal{Z}_{h_1}') \\
&+ P(Y = 1 | a_1, z \in \mathcal{Z}_{h_1,0}') \cdot P(S = 0 | a_1, z \in \mathcal{Z}_{h_1}') \\
&= P(Y = 1 | a_2, z \in \mathcal{Z}_{h_2,1}') \cdot P(S = 1 | a_2, z \in \mathcal{Z}_{h_2}') \\
&+ P(Y = 1 | a_2, z \in \mathcal{Z}_{h_2,0}') \cdot P(S = 0 | a_2, z \in \mathcal{Z}_{h_2}') + \alpha^*.
\end{aligned}
\tag{36}
$$

*We can quantify the utility disparity gap under different confidence settings when the decision-maker consistently chooses $T = 1$ as follows:*

$$
\begin{aligned}
&\left( \mathbb{E}\left[u(1, Y) | a_1, z \in \mathcal{Z}_{h_1,1}'\right] - \mathbb{E}\left[u(1, Y) | a_1, z \in \mathcal{Z}_{h_1,0}'\right] \right) \\
&- \left( \mathbb{E}\left[u(1, Y) | a_2, z \in \mathcal{Z}_{h_2,1}'\right] - \mathbb{E}\left[u(1, Y) | a_2, z \in \mathcal{Z}_{h_2,0}'\right] \right) \\
&= \left( u(1, 1) - u(1, 0) \right) \cdot \Delta_1.
\end{aligned}
\tag{37}
$$

*where,*

$$
\begin{aligned}
\Delta_1 = &P(Y = 1 | a_1, z \in \mathcal{Z}_{h_1,1}') - P(Y = 1 | a_1, z \in \mathcal{Z}_{h_1,0}') \\
&- P(Y = 1 | a_2, z \in \mathcal{Z}_{h_2,1}') + P(Y = 1 | a_2, z \in \mathcal{Z}_{h_2,0}').
\end{aligned}
$$

*Similarly, we can define the utility disparity gap under different confidence settings when the decision-maker consistently chooses $T = 0$,*

$$
\begin{aligned}
&\left( \mathbb{E}\left[u(0, Y) | a_1, z \in \mathcal{Z}_{h_1,1}'\right] - \mathbb{E}\left[u(0, Y) | a_1, z \in \mathcal{Z}_{h_1,0}'\right] \right) \\
&- \left( \mathbb{E}\left[u(0, Y) | a_2, z \in \mathcal{Z}_{h_2,1}'\right] - \mathbb{E}\left[u(0, Y) | a_2, z \in \mathcal{Z}_{h_2,0}'\right] \right) \\
&= (u(0, 1) - u(0, 0)) \cdot \Delta_1.
\end{aligned}
\tag{38}
$$

*As $P(Y = 1 \mid f_A(z) = a_1, z \in \mathcal{Z}_{h_1,1}) - P(Y = 1 \mid f_A(z) = a_1, z \in \mathcal{Z}_{h_1,0}) \leq 0$, according to Lemma A.2, we have:*

$$
\begin{aligned}
&\mathbb{E}\left[u(1, Y) | f_A(z) = a_1, z \in \mathcal{Z}_{h_1,1}\right] - \mathbb{E}\left[u(1, Y) | f_A(z) = a_1, z \in \mathcal{Z}_{h_1,0}\right] \\
&\leq 0 \leq \mathbb{E}\left[u(0, Y) | f_A(z) = a_1, z \in \mathcal{Z}_{h_1,1}\right] - \mathbb{E}\left[u(0, Y) | f_A(z) = a_1, z \in \mathcal{Z}_{h_1,0}\right].
\end{aligned}
\tag{39}
$$

*Combining Eqs. (37), (38) and (39), it holds that,*

$$
\begin{aligned}
&\left( \mathbb{E}\left[u(1, Y) | a_2, z \in \mathcal{Z}_{h_2,1}'\right] - \mathbb{E}\left[u(1, Y) | a_2, z \in \mathcal{Z}_{h_2,0}'\right] \right) \\
&- \left( \mathbb{E}\left[u(0, Y) | a_2, z \in \mathcal{Z}_{h_2,1}'\right] - \mathbb{E}\left[u(0, Y) | a_2, z \in \mathcal{Z}_{h_2,0}'\right] \right) \\
&\leq \left( u(0, 1) - u(0, 0) - u(1, 1) + u(1, 0) \right) \cdot \Delta_1.
\end{aligned}
\tag{40}
$$

*As $P(Y = 1 \mid a_2, z \in \mathcal{Z}_{h_2,1}) - P(Y = 1 \mid a_2, z \in \mathcal{Z}_{h_2,0}) > 0$, it holds that,*

$$
\begin{aligned}
&\mathbb{E}\left[u(1, Y) | a_2, z \in \mathcal{Z}_{h_2,1}\right] - \mathbb{E}\left[u(1, Y) | a_2, z \in \mathcal{Z}_{h_2,0}\right] \\
&> 0 \geq \mathbb{E}\left[u(0, Y) | a_2, z \in \mathcal{Z}_{h_2,1}\right] - \mathbb{E}\left[u(0, Y) | a_2, z \in \mathcal{Z}_{h_2,0}\right].
\end{aligned}
\tag{41}
$$

*Based on Eq. (41), the upper bound of the utility disparity of policy $\pi$ is,*

$$
\begin{aligned}
0 \leq &\left| \mathbb{E}_\pi \left[ u(T, Y) | a_2, z \in \mathcal{Z}_{h_2,1} \right] - \mathbb{E}_\pi \left[ u(T, Y) | a_2, z \in \mathcal{Z}_{h_2,0} \right] \right| \\
\leq &\left( \mathbb{E} \left[ u(1, Y) | a_2, z \in \mathcal{Z}_{h_2,1} \right] - \mathbb{E} \left[ u(1, Y) | a_2, z \in \mathcal{Z}_{h_2,0} \right] \right) \\
&- \left( \mathbb{E} \left[ u(0, Y) | a_2, z \in \mathcal{Z}_{h_2,1} \right] - \mathbb{E} \left[ u(0, Y) | a_2, z \in \mathcal{Z}_{h_2,0} \right] \right) \\
= &\left( 1 - \frac{\alpha}{2} \right) \\
&\cdot \left( \left( \mathbb{E} \left[ u(1, Y) | a_2, z \in \mathcal{Z}'_{h_2,1} \right] - \mathbb{E} \left[ u(1, Y) | a_2, z \in \mathcal{Z}'_{h_2,0} \right] \right) \right. \\
&- \left. \left( \mathbb{E} \left[ u(0, Y) | a_2, z \in \mathcal{Z}'_{h_2,1} \right] - \mathbb{E} \left[ u(0, Y) | a_2, z \in \mathcal{Z}'_{h_2,0} \right] \right) \right) \\
&+ \frac{\alpha}{2} \\
&\cdot \left( \left( \mathbb{E} \left[ u(1, Y) | a_2, z \in \mathcal{Z}_{h_2,1} \setminus \mathcal{Z}'_{h_2,1} \right] - \mathbb{E} \left[ u(1, Y) | a_2, z \in \mathcal{Z}_{h_2,0} \setminus \mathcal{Z}'_{h_2,0} \right] \right) \right. \\
&- \left. \left( \mathbb{E} \left[ u(0, Y) | a_2, z \in \mathcal{Z}_{h_2,1} \setminus \mathcal{Z}'_{h_2,1} \right] - \mathbb{E} \left[ u(0, Y) | a_2, z \in \mathcal{Z}_{h_2,0} \setminus \mathcal{Z}'_{h_2,0} \right] \right) \right).
\end{aligned}
\tag{42}
$$

*For $z \in \mathcal{Z}_{h_2} \setminus \mathcal{Z}'_{h_2}$, as*

$$
\begin{aligned}
u(1, 0) &\leq \mathbb{E} \left[ u(1, Y) \mid a_2, z \in \mathcal{Z}_{h_2} \setminus \mathcal{Z}'_{h_2} \right] \leq u(1, 1), \\
u(0, 1) &\leq \mathbb{E} \left[ u(0, Y) \mid a_2, z \in \mathcal{Z}_{h_2} \setminus \mathcal{Z}'_{h_2} \right] \leq u(0, 0).
\end{aligned}
\tag{43}
$$

*we have:*

$$
0 < \mathbb{E} \left[ u(1, Y) \mid a_2, z \in \mathcal{Z}_{h_2,1} \setminus \mathcal{Z}'_{h_2,1} \right] - \mathbb{E} \left[ u(1, Y) \mid a_2, z \in \mathcal{Z}_{h_2,0} \setminus \mathcal{Z}'_{h_2,0} \right] \leq u(1, 1) - u(1, 0),
\tag{44}
$$

$$
u(0, 1) - u(0, 0) \leq \mathbb{E} \left[ u(0, Y) \mid a_2, z \in \mathcal{Z}_{h_2,1} \setminus \mathcal{Z}'_{h_2,1} \right] - \mathbb{E} \left[ u(0, Y) \mid a_2, z \in \mathcal{Z}_{h_2,0} \setminus \mathcal{Z}'_{h_2,0} \right] \leq 0.
\tag{45}
$$

*Then, Eq. (42) can be reorganized as follows:*

$$
\begin{aligned}
0 \leq &\left| \mathbb{E}_\pi \left[ u(T, Y) | a_2, z \in \mathcal{Z}_{h_2,1} \right] - \mathbb{E}_\pi \left[ u(T, Y) | a_2, z \in \mathcal{Z}_{h_2,0} \right] \right| \\
\leq &\left( 1 - \frac{\alpha}{2} \right) \cdot \left( u(1, 0) - u(1, 1) - u(0, 0) + u(0, 1) \right) \cdot \Delta_1 \\
&+ \frac{\alpha}{2} \cdot \left( u(1, 1) - u(1, 0) - u(0, 1) + u(0, 0) \right).
\end{aligned}
\tag{46}
$$

*Based on Eq. (33) and (34), it follows that:*

$$
\Delta_1 \leq 0.
\tag{47}
$$

*Therefore, the optimal utility disparity is achieved when:*

$$
\Delta_1 \equiv 0^-.
\tag{48}
$$

*When $f_A$ is perfectly aligned with $f_H$ ($\alpha^* \leq \alpha = 0$) and Eq. (48) does not hold, there are infinitely many cases that*

$$
\begin{aligned}
&\left| \mathbb{E}_\pi \left[ u(T, Y) | a_2, z \in \mathcal{Z}_{h_2,1} \right] - \mathbb{E}_\pi \left[ u(T, Y) | a_2, z \in \mathcal{Z}_{h_2,0} \right] \right| \\
&> \left| \mathbb{E}_{\pi^*} \left[ u(T, Y) | a_2, z \in \mathcal{Z}_{h_2,1} \right] - \mathbb{E}_{\pi^*} \left[ u(T, Y) | a_2, z \in \mathcal{Z}_{h_2,0} \right] \right|.
\end{aligned}
\tag{49}
$$

**Case 2.** *For any confidence $[h_1, a_1]$ with $a_1 > \bar{a}_1$, according to Eq. (32), it holds that,*

$$
P(Y = 1 \mid a_1, z \in \mathcal{Z}_{h_1,1}) - P(Y = 1 \mid a_1, z \in \mathcal{Z}_{h_1,0}) > 0.
\tag{50}
$$

*Furthermore, there exists another $[h_2, a_2]$, where $\bar{a}_2 > a_2 > a_1$ and $h_2 > h_1$ such that,*

$$
P(Y = 1 \mid a_2, z \in \mathcal{Z}_{h_2,1}) - P(Y = 1 \mid a_2, z \in \mathcal{Z}_{h_2,0}) \leq 0.
\tag{51}
$$

*According to Lemma A.2, we have:*

$$
\begin{aligned}
&\mathbb{E} \left[ u(1, Y) | a_1, z \in \mathcal{Z}_{h_1,1} \right] - \mathbb{E} \left[ u(1, Y) | a, z \in \mathcal{Z}_{h,0} \right] \\
&> \mathbb{E} \left[ u(0, Y) | a_1, z \in \mathcal{Z}_{h_1,1} \right] - \mathbb{E} \left[ u(0, Y) | a, z \in \mathcal{Z}_{h,0} \right].
\end{aligned}
\tag{52}
$$

*Combining Eqs. (52), (38) and (39), it holds that,*

$$
\begin{aligned}
&\left( \mathbb{E}\left[ u(0,Y)|a_2, z \in \mathcal{Z}'_{h_2,1} \right] - \mathbb{E}\left[ u(0,Y)|a_2, z \in \mathcal{Z}'_{h_2,0} \right] \right) \\
&- \left( \mathbb{E}\left[ u(1,Y)|a_2, z \in \mathcal{Z}'_{h_2,1} \right] - \mathbb{E}\left[ u(1,Y)|a_2, z \in \mathcal{Z}'_{h_2,0} \right] \right) \\
&\leq (u(1,1) - u(1,0) - u(0,1) + u(0,0)) \cdot \Delta_2.
\end{aligned}
\tag{53}
$$

*where,*

$$
\begin{aligned}
\Delta_2 = {}& P(Y = 1|a_1, z \in \mathcal{Z}'_{h_1,1}) - P(Y = 1|a_1, z \in \mathcal{Z}'_{h_1,0}) \\
& - P(Y = 1|a_2, z \in \mathcal{Z}'_{h_2,1}) + P(Y = 1|a_2, z \in \mathcal{Z}'_{h_2,0})
\end{aligned}
$$

*As $P(Y = 1 \mid a_2, z \in \mathcal{Z}_{h_2,1}) - P(Y = 1 \mid a_2, z \in \mathcal{Z}_{h_2,0}) \leq 0$, it holds that,*

$$
\begin{aligned}
&\mathbb{E}\left[ u(1,Y)|a_2, z \in \mathcal{Z}_{h_2,1} \right] - \mathbb{E}\left[ u(1,Y)|a_2, z \in \mathcal{Z}_{h_2,0} \right] \\
&\leq 0 \leq \mathbb{E}\left[ u(0,Y)|a_2, z \in \mathcal{Z}_{h_2,1} \right] - \mathbb{E}\left[ u(0,Y)|a_2, z \in \mathcal{Z}_{h_2,0} \right].
\end{aligned}
\tag{54}
$$

*Based on Eq. (54), the upper bound of the utility disparity of policy $\pi$ is,*

$$
\begin{aligned}
0 \leq {}& |\mathbb{E}_\pi\left[ u(T,Y)|a_2, z \in \mathcal{Z}_{h_2,1} \right] - \mathbb{E}_\pi\left[ u(T,Y)|a_2, z \in \mathcal{Z}_{h_2,0} \right]| \\
\leq {}& (\mathbb{E}\left[ u(0,Y)|a_2, z \in \mathcal{Z}_{h_2,1} \right] - \mathbb{E}\left[ u(0,Y)|a_2, z \in \mathcal{Z}_{h_2,0} \right]) \\
& - (\mathbb{E}\left[ u(1,Y)|a_2, z \in \mathcal{Z}_{h_2,1} \right] - \mathbb{E}\left[ u(1,Y)|a_2, z \in \mathcal{Z}_{h_2,0} \right]) \\
= {}& \left( 1 - \frac{\alpha}{2} \right) \\
& \cdot \left( \left( \mathbb{E}\left[ u(0,Y)|a_2, z \in \mathcal{Z}'_{h_2,1} \right] - \mathbb{E}\left[ u(0,Y)|a_2, z \in \mathcal{Z}'_{h_2,0} \right] \right) \right. \\
& \left. - \left( \mathbb{E}\left[ u(1,Y)|a_2, z \in \mathcal{Z}'_{h_2,1} \right] - \mathbb{E}\left[ u(1,Y)|a_2, z \in \mathcal{Z}'_{h_2,0} \right] \right) \right) \\
& + \frac{\alpha}{2} \\
& \cdot \left( \left( \mathbb{E}\left[ u(0,Y)|a_2, z \in \mathcal{Z}_{h_2,1} \setminus \mathcal{Z}'_{h_2,1} \right] - \mathbb{E}\left[ u(0,Y)|a_2, z \in \mathcal{Z}_{h_2,0} \setminus \mathcal{Z}'_{h_2,0} \right] \right) \right. \\
& \left. - \left( \mathbb{E}\left[ u(1,Y)|a_2, z \in \mathcal{Z}_{h_2,1} \setminus \mathcal{Z}'_{h_2,1} \right] - \mathbb{E}\left[ u(1,Y)|a_2, z \in \mathcal{Z}_{h_2,0} \setminus \mathcal{Z}'_{h_2,0} \right] \right) \right).
\end{aligned}
\tag{55}
$$

*For $z \in \mathcal{Z}_{h_2} \setminus \mathcal{Z}'_{h_2}$, according to Eq. (43), it holds that,*

$$
u(1,0) - u(1,1) \leq \mathbb{E}\left[ u(1,Y) \mid a_2, z \in \mathcal{Z}_{h_2,1} \setminus \mathcal{Z}'_{h_2,1} \right] - \mathbb{E}\left[ u(1,Y) \mid a_2, z \in \mathcal{Z}_{h_2,0} \setminus \mathcal{Z}'_{h_2,0} \right] \leq 0,
\tag{56}
$$

$$
0 \leq \mathbb{E}\left[ u(0,Y) \mid a_2, z \in \mathcal{Z}_{h_2,1} \setminus \mathcal{Z}'_{h_2,1} \right] - \mathbb{E}\left[ u(0,Y) \mid a_2, z \in \mathcal{Z}_{h_2,0} \setminus \mathcal{Z}'_{h_2,0} \right] \leq u(0,0) - u(0,1).
\tag{57}
$$

*Then, Eq. (55) can be reorganized as follows:*

$$
\begin{aligned}
0 \leq {}& |\mathbb{E}_\pi\left[ u(T,Y)|a_2, z \in \mathcal{Z}_{h_2,1} \right] - \mathbb{E}_\pi\left[ u(T,Y)|a_2, z \in \mathcal{Z}_{h_2,0} \right]| \\
\leq {}& \left( 1 - \frac{\alpha}{2} \right) \cdot (u(1,1) - u(1,0) - u(0,1) + u(0,0)) \cdot \Delta_2 \\
& + \frac{\alpha}{2} \cdot (u(1,1) - u(1,0) - u(0,1) + u(0,0)).
\end{aligned}
\tag{58}
$$

*Based on Eq. (50) and (51), it follows that:*

$$
\Delta_2 \geq 0.
\tag{59}
$$

*Therefore, the optimal utility disparity is achieved when:*

$$
\Delta_2 \equiv 0^+.
\tag{60}
$$

*When $f_A$ is perfectly aligned with $f_H$ ($\alpha^* \leq \alpha = 0$) and Eq. (60) does not hold, there are infinitely many cases that*

$$
\begin{aligned}
&|\mathbb{E}_\pi\left[ u(T,Y)|a_2, z \in \mathcal{Z}_{h_2,1} \right] - \mathbb{E}_\pi\left[ u(T,Y)|a_2, z \in \mathcal{Z}_{h_2,0} \right]| \\
&> |\mathbb{E}_{\pi^*}\left[ u(T,Y)|a_2, z \in \mathcal{Z}_{h_2,1} \right] - \mathbb{E}_{\pi^*}\left[ u(T,Y)|a_2, z \in \mathcal{Z}_{h_2,0} \right]|.
\end{aligned}
\tag{61}
$$

**Case 3.** *For any confidence $[h_1, a_1]$ with $a_1 < \bar{a}_1$, according to Eq. (32), it holds that,*

$$P(Y = 1 \mid a_1, z \in \mathcal{Z}_{h_1,1}) - P(Y = 1 \mid a_1, z \in \mathcal{Z}_{h_1,0}) \leq 0. \tag{62}$$

*Furthermore, there exists another $[h_2, a_2]$, where $\bar{a}_2 > a_2 > a_1$ and $h_2 > h_1$ such that,*

$$P(Y = 1 \mid a_2, z \in \mathcal{Z}_{h_2,1}) - P(Y = 1 \mid a_2, z \in \mathcal{Z}_{h_2,0}) \leq 0. \tag{63}$$

*When the decision-maker consistently chooses $T = 1$, it holds that:*

$$
\begin{aligned}
&\left( \mathbb{E}\left[ u(1, Y) | a_1, z \in \mathcal{Z}'_{h_1,1} \right] - \mathbb{E}\left[ u(1, Y) | a_1, z \in \mathcal{Z}'_{h_1,0} \right] \right) \\
&+ \left( \mathbb{E}\left[ u(1, Y) | a_2, z \in \mathcal{Z}'_{h_2,1} \right] - \mathbb{E}\left[ u(1, Y) | a_2, z \in \mathcal{Z}'_{h_2,0} \right] \right) \\
&= (u(1,1) - u(1,0)) \cdot \Delta_3.
\end{aligned}
\tag{64}
$$

*where,*

$$
\begin{aligned}
\Delta_3 = {}& P(Y = 1 | a_1, z \in \mathcal{Z}'_{h_1,1}) - P(Y = 1 | a_1, z \in \mathcal{Z}'_{h_1,0}) \\
&+ P(Y = 1 | a_2, z \in \mathcal{Z}'_{h_2,1}) - P(Y = 1 | a_2, z \in \mathcal{Z}'_{h_2,0}).
\end{aligned}
$$

*Similarly, when the decision-maker consistently chooses $T = 0$, it holds that,*

$$
\begin{aligned}
&\left( \mathbb{E}\left[ u(0, Y) | a_1, z \in \mathcal{Z}'_{h_1,1} \right] - \mathbb{E}\left[ u(0, Y) | a, z \in \mathcal{Z}'_{h_1,0} \right] \right) \\
&+ \left( \mathbb{E}\left[ u(0, Y) | a_2, z \in \mathcal{Z}'_{h_2,1} \right] - \mathbb{E}\left[ u(0, Y) | a_2, z \in \mathcal{Z}'_{h_2,0} \right] \right) \\
&= (u(0,1) - u(0,0)) \cdot \Delta_3.
\end{aligned}
\tag{65}
$$

*According to Corollary A.2, we have:*

$$
\begin{aligned}
&\mathbb{E}\left[ u(1, Y) | a_1, z \in \mathcal{Z}_{h_1,1} \right] - \mathbb{E}\left[ u(1, Y) | a_1, z \in \mathcal{Z}_{h_1,0} \right] \\
&\leq \mathbb{E}\left[ u(0, Y) | a_1, z \in \mathcal{Z}_{h_1,1} \right] - \mathbb{E}\left[ u(0, Y) | a_1, z \in \mathcal{Z}_{h_1,0} \right].
\end{aligned}
\tag{66}
$$

*Combining Eqs. (66), (64) and (65), it holds that,*

$$
\begin{aligned}
&\left( \mathbb{E}\left[ u(0, Y) | a_2, z \in \mathcal{Z}'_{h_2,1} \right] - \mathbb{E}\left[ u(0, Y) | a_2, z \in \mathcal{Z}'_{h_2,0} \right] \right) \\
&- \left( \mathbb{E}\left[ u(1, Y) | a_2, z \in \mathcal{Z}'_{h_2,1} \right] - \mathbb{E}\left[ u(1, Y) | a_2, z \in \mathcal{Z}'_{h_2,0} \right] \right) \\
&\leq (u(0,1) - u(0,0) - u(1,1) + u(1,0)) \cdot \Delta_3.
\end{aligned}
\tag{67}
$$

*As $P(Y = 1 \mid a_2, z \in \mathcal{Z}_{h_2,1}) - P(Y = 1 \mid a_2, z \in \mathcal{Z}_{h_2,0}) \leq 0$, it holds that,*

$$
\begin{aligned}
&\mathbb{E}\left[ u(1, Y) | a_2, z \in \mathcal{Z}_{h_2,1} \right] - \mathbb{E}\left[ u(1, Y) | a_2, z \in \mathcal{Z}_{h_2,0} \right] \\
&\leq 0 \leq \mathbb{E}\left[ u(0, Y) | a_2, z \in \mathcal{Z}_{h_2,1} \right] - \mathbb{E}\left[ u(0, Y) | a_2, z \in \mathcal{Z}_{h_2,0} \right].
\end{aligned}
\tag{68}
$$

*Based on Eq. (68), the upper bound of the utility disparity of policy $\pi$ is,*

$$
\begin{aligned}
0 \leq {}& |\mathbb{E}_\pi\left[ u(T, Y) | a_2, z \in \mathcal{Z}_{h_2,1} \right] - \mathbb{E}_\pi\left[ u(T, Y) | a_2, z \in \mathcal{Z}_{h_2,0} \right]| \\
\leq {}& \left( \mathbb{E}\left[ u(0, Y) | a_2, z \in \mathcal{Z}_{h_2,1} \right] - \mathbb{E}\left[ u(0, Y) | a_2, z \in \mathcal{Z}_{h_2,0} \right] \right) \\
&- \left( \mathbb{E}\left[ u(1, Y) | a_2, z \in \mathcal{Z}_{h_2,1} \right] - \mathbb{E}\left[ u(1, Y) | a_2, z \in \mathcal{Z}_{h_2,0} \right] \right) \\
= {}& \left( 1 - \frac{\alpha}{2} \right) \\
&\cdot \Big( \left( \mathbb{E}\left[ u(0, Y) | a_2, z \in \mathcal{Z}'_{h_2,1} \right] - \mathbb{E}\left[ u(0, Y) | a_2, z \in \mathcal{Z}'_{h_2,0} \right] \right) \\
&- \left( \mathbb{E}\left[ u(1, Y) | a_2, z \in \mathcal{Z}'_{h_2,1} \right] - \mathbb{E}\left[ u(1, Y) | a_2, z \in \mathcal{Z}'_{h_2,0} \right] \right) \Big) \\
&+ \frac{\alpha}{2} \\
&\cdot \Big( \left( \mathbb{E}\left[ u(0, Y) | a_2, z \in \mathcal{Z}_{h_2,1} \setminus \mathcal{Z}'_{h_2,1} \right] - \mathbb{E}\left[ u(0, Y) | a_2, z \in \mathcal{Z}_{h_2,0} \setminus \mathcal{Z}'_{h_2,0} \right] \right) \\
&- \left( \mathbb{E}\left[ u(1, Y) | a_2, z \in \mathcal{Z}_{h_2,1} \setminus \mathcal{Z}'_{h_2,1} \right] - \mathbb{E}\left[ u(1, Y) | a_2, z \in \mathcal{Z}_{h_2,0} \setminus \mathcal{Z}'_{h_2,0} \right] \right) \Big).
\end{aligned}
\tag{69}
$$

*For $z \in \mathcal{Z}_{h_2} \setminus \mathcal{Z}'_{h_2}$, according to Eq. (43), it holds that,*

$$u(1,0) - u(1,1) \leq \mathbb{E}\left[ u(1, Y) \mid a_2, z \in \mathcal{Z}_{h_2,1} \setminus \mathcal{Z}'_{h_2,1} \right] - \mathbb{E}\left[ u(1, Y) \mid a_2, z \in \mathcal{Z}_{h_2,0} \setminus \mathcal{Z}'_{h_2,0} \right] \leq 0. \tag{70}$$

$$0 \leq \mathbb{E}\left[u(0,Y) \mid a_2, z \in \mathcal{Z}_{h_2,1} \setminus \mathcal{Z}'_{h_2,1}\right] - \mathbb{E}\left[u(0,Y) \mid a_2, z \in \mathcal{Z}_{h_2,0} \setminus \mathcal{Z}'_{h_2,0}\right] \leq u(0,0) - u(0,1). \tag{71}$$

*Then, Eq. (69) can be reorganized as follows:*

$$0 \leq |\mathbb{E}_\pi\left[u(T,Y)|a_2, z \in \mathcal{Z}_{h_2,1}\right] - \mathbb{E}_\pi\left[u(T,Y)|a_2, z \in \mathcal{Z}_{h_2,0}\right]|$$
$$\leq \left(1 - \frac{\alpha}{2}\right) \cdot (u(1,0) - u(1,1) - u(0,0) + u(0,1)) \cdot \Delta_3 \tag{72}$$
$$+ \frac{\alpha}{2} \cdot (u(1,1) - u(1,0) - u(0,1) + u(0,0)).$$

*Based on Eq. (62) and (63), it follows that:*

$$\Delta_3 \leq 0. \tag{73}$$

*Therefore, the optimal utility disparity is achieved when:*

$$\Delta_3 \equiv 0^-. \tag{74}$$

*When $f_A$ is perfectly aligned with $f_H$ ($\alpha^* \leq \alpha = 0$) and Eq. (74) does not hold, there are infinitely many cases that*

$$|\mathbb{E}_\pi\left[u(T,Y)|a_2, z \in \mathcal{Z}_{h_2,1}\right] - \mathbb{E}_\pi\left[u(T,Y)|a_2, z \in \mathcal{Z}_{h_2,0}\right]|$$
$$> |\mathbb{E}_{\pi^*}\left[u(T,Y)|a_2, z \in \mathcal{Z}_{h_2,1}\right] - \mathbb{E}_{\pi^*}\left[u(T,Y)|a_2, z \in \mathcal{Z}_{h_2,0}\right]|. \tag{75}$$

**Case 4.** *For any confidence $[h_1, a_1]$ with $a_1 > \bar{a}_1$, according to Eq. (32), it holds that,*

$$P(Y = 1 \mid a_1, z \in \mathcal{Z}_{h_1,1}) - P(Y = 1 \mid a_1, z \in \mathcal{Z}_{h_1,0}) > 0. \tag{76}$$

*Furthermore, there exists another $[h_2, a_2]$, where $a_2 > a_1 > \bar{a}_2$ and $h_2 > h_1$ such that,*

$$P(Y = 1 \mid a_2, z \in \mathcal{Z}_{h_2,1}) - P(Y = 1 \mid a_2, z \in \mathcal{Z}_{h_2,0}) > 0. \tag{77}$$

*According to Lemma A.2, we have:*

$$\mathbb{E}\left[u(1,Y)|a_1, z \in \mathcal{Z}_{h_1,1}\right] - \mathbb{E}\left[u(1,Y)|a_1, z \in \mathcal{Z}_{h_1,0}\right]$$
$$> \mathbb{E}\left[u(0,Y)|a_1, z \in \mathcal{Z}_{h_1,1}\right] - \mathbb{E}\left[u(0,Y)|a_1, z \in \mathcal{Z}_{h_1,0}\right]. \tag{78}$$

*Combining Eqs. (78), (64) and (65), it holds that,*

$$\left(\mathbb{E}\left[u(1,Y)|a_2, z \in \mathcal{Z}'_{h_2,1}\right] - \mathbb{E}\left[u(1,Y)|a_2, z \in \mathcal{Z}'_{h_2,0}\right]\right)$$
$$- \left(\mathbb{E}\left[u(0,Y)|a_2, z \in \mathcal{Z}'_{h_2,1}\right] - \mathbb{E}\left[u(0,Y)|a_2, z \in \mathcal{Z}'_{h_2,0}\right]\right) \tag{79}$$
$$\leq (u(1,1) - u(1,0) - u(0,1) + u(0,0)) \cdot \Delta_4.$$

*where,*

$$\Delta_4 = P(Y = 1|a_1, z \in \mathcal{Z}'_{h_1,1}) - P(Y = 1|a_1, z \in \mathcal{Z}'_{h_1,0})$$
$$+ P(Y = 1|a_2, z \in \mathcal{Z}'_{h_2,1}) - P(Y = 1|a_2, z \in \mathcal{Z}'_{h_2,0}).$$

*As $P(Y = 1 \mid a_2, z \in \mathcal{Z}_{h_2,1}) - P(Y = 1 \mid a_2, z \in \mathcal{Z}_{h_2,0}) > 0$, it holds that,*

$$\mathbb{E}\left[u(0,Y)|a_2, z \in \mathcal{Z}_{h_2,1}\right] - \mathbb{E}\left[u(0,Y)|a_2, z \in \mathcal{Z}_{h_2,0}\right]$$
$$\leq 0 < \mathbb{E}\left[u(1,Y)|a_2, z \in \mathcal{Z}_{h_2,1}\right] - \mathbb{E}\left[u(1,Y)|a_2, z \in \mathcal{Z}_{h_2,0}\right]. \tag{80}$$

*Based on Eq. (80), the upper bound of the utility disparity of policy $\pi$ is,*

$$0 \leq |\mathbb{E}_\pi\left[u(T,Y)|a_2, z \in \mathcal{Z}_{h_2,1}\right] - \mathbb{E}_\pi\left[u(T,Y)|a_2, z \in \mathcal{Z}_{h_2,0}\right]|$$
$$\leq \left(\mathbb{E}\left[u(1,Y)|a_2, z \in \mathcal{Z}_{h_2,1}\right] - \mathbb{E}\left[u(1,Y)|a_2, z \in \mathcal{Z}_{h_2,0}\right]\right)$$
$$- \left(\mathbb{E}\left[u(0,Y)|a_2, z \in \mathcal{Z}_{h_2,1}\right] - \mathbb{E}\left[u(0,Y)|a_2, z \in \mathcal{Z}_{h_2,0}\right]\right)$$
$$= \left(1 - \frac{\alpha}{2}\right)$$
$$\cdot \left(\left(\mathbb{E}\left[u(1,Y)|a_2, z \in \mathcal{Z}'_{h_2,1}\right] - \mathbb{E}\left[u(1,Y)|a_2, z \in \mathcal{Z}'_{h_2,0}\right]\right)\right.$$
$$\left. - \left(\mathbb{E}\left[u(0,Y)|a_2, z \in \mathcal{Z}'_{h_2,1}\right] - \mathbb{E}\left[u(0,Y)|a_2, z \in \mathcal{Z}'_{h_2,0}\right]\right)\right) \tag{81}$$
$$+ \frac{\alpha}{2}$$
$$\cdot \left(\left(\mathbb{E}\left[u(1,Y)|a_2, z \in \mathcal{Z}_{h_2,1} \setminus \mathcal{Z}'_{h_2,1}\right] - \mathbb{E}\left[u(1,Y)|a_2, z \in \mathcal{Z}_{h_2,0} \setminus \mathcal{Z}'_{h_2,0}\right]\right)\right.$$
$$\left. - \left(\mathbb{E}\left[u(0,Y)|a_2, z \in \mathcal{Z}_{h_2,1} \setminus \mathcal{Z}'_{h_2,1}\right] - \mathbb{E}\left[u(0,Y)|a_2, z \in \mathcal{Z}_{h_2,0} \setminus \mathcal{Z}'_{h_2,0}\right]\right)\right).$$

*For $z \in \mathcal{Z}_{h_2} \setminus \mathcal{Z}'_{h_2}$, according to Eq. (43), it holds that,*

$$0 \le \mathbb{E}\left[u(1,Y) \mid a_2, z \in \mathcal{Z}_{h_2,1} \setminus \mathcal{Z}'_{h_2,1}\right] - \mathbb{E}\left[u(1,Y) \mid a_2, z \in \mathcal{Z}_{h_2,0} \setminus \mathcal{Z}'_{h_2,0}\right] \le u(1,1) - u(1,0), \tag{82}$$

$$u(0,1) - u(0,0) \le \mathbb{E}\left[u(0,Y) \mid a_2, z \in \mathcal{Z}_{h_2,1} \setminus \mathcal{Z}'_{h_2,1}\right] - \mathbb{E}\left[u(0,Y) \mid a_2, z \in \mathcal{Z}_{h_2,0} \setminus \mathcal{Z}'_{h_2,0}\right] \le 0. \tag{83}$$

*Then, Eq. (81) can be reorganized as follows:*

$$\begin{aligned}
0 &\le \left|\mathbb{E}_\pi\left[u(T,Y)|a_2, z \in \mathcal{Z}_{h_2,1}\right] - \mathbb{E}_\pi\left[u(T,Y)|a_2, z \in \mathcal{Z}_{h_2,0}\right]\right| \\
&\le \left(1 - \frac{\alpha}{2}\right) \cdot (u(1,1) - u(1,0) - u(0,1) + u(0,0)) \cdot \Delta_4 \\
&\quad + \frac{\alpha}{2} \cdot (u(1,1) - u(1,0) - u(0,1) + u(0,0)).
\end{aligned} \tag{84}$$

*Based on Eq. (76) and (77), it follows that:*

$$\Delta_4 > 0. \tag{85}$$

*Therefore, the optimal utility disparity is achieved when:*

$$\Delta_4 \equiv 0^+. \tag{86}$$

*When $f_A$ is perfectly aligned with $f_H$ ($\alpha^* \le \alpha = 0$) and Eq. (86) does not hold, there are infinitely many cases that*

$$\begin{aligned}
&\left|\mathbb{E}_\pi\left[u(T,Y)|a_2, z \in \mathcal{Z}'_{h_2,1}\right] - \mathbb{E}_\pi\left[u(T,Y)|a_2, z \in \mathcal{Z}'_{h_2,0}\right]\right| \\
&> \left|\mathbb{E}_{\pi^*}\left[u(T,Y)|a_2, z \in \mathcal{Z}'_{h_2,1}\right] - \mathbb{E}_{\pi^*}\left[u(T,Y)|a_2, z \in \mathcal{Z}'_{h_2,0}\right]\right|.
\end{aligned} \tag{87}$$

*Based on the above proof, we have demonstrated the existence of scenarios in which, even when $f_A$ is perfectly aligned with $f_H$, any monotone policy $\pi$ leads to a suboptimal utility disparity, thereby supporting Theorem 3.4.* $\qquad\square$

## A.5 PROOF OF THEOREM 3.6

**Theorem 3.6** (Utility disparity upper bound under $\alpha$-human-alignment) For a given AI-assisted decision-making process with a utility function $u(T,Y)$ satisfying Eq. (1) and the human decision-maker with any monotone AI-assisted decision policy $\pi \in \Pi(H, A)$, if the AI confidence function $f_A$ is $\alpha_h$-human-alignment and satisfies $\alpha_g$-inter-group-alignment, then the utility disparity is bounded by,

$$\begin{aligned}
&\left|\mathbb{E}_\pi\left[u(T,Y)|f_A(z) = a, z \in \mathcal{Z}_{h,1}\right] - \mathbb{E}_\pi\left[u(T,Y)|f_A(z) = a, z \in \mathcal{Z}_{h,0}\right]\right| \\
&\le (u(1,1) - u(0,1) - u(1,0) + u(0,0)) \cdot \left(\frac{\alpha_h}{2} + \left(1 - \frac{\alpha_h}{2}\right) \cdot \left(3\alpha_g - \alpha_g^2\right)\right).
\end{aligned} \tag{88}$$

**Proof.** *Given $\alpha_g$-inter-group-alignment, for any two confidence levels $\{h_1, a_1\}$ and $\{h_2, a_2\}$ with $a_2 > a_1$ and $h_2 > h_1$, the following conditions hold for all $z \in \mathcal{Z}''_{h_1}$ and $z \in \mathcal{Z}''_{h_2}$, respectively:*

$$-\alpha_g \le P(Y = 1|a_1, z \in \mathcal{Z}''_{h_1,1}) - P(Y = 1|a_1, z \in \mathcal{Z}''_{h_1,0}) \le \alpha_g. \tag{89}$$

$$-\alpha_g \le P(Y = 1|a_2, z \in \mathcal{Z}''_{h_2,1}) - P(Y = 1|a_2, z \in \mathcal{Z}''_{h_2,0}) \le \alpha_g, \tag{90}$$

*where $\mathcal{Z}''_{h_1} \subset \mathcal{Z}_{h_1}$ and $\mathcal{Z}''_{h_2} \subset \mathcal{Z}_{h_2}$ with $\left|\mathcal{Z}''_{h_1}\right| \ge (1 - \alpha_g/2) \cdot |\mathcal{Z}_h|$ and $\left|\mathcal{Z}''_{h_2}\right| \ge (1 - \alpha_g/2) \cdot |\mathcal{Z}_{h_2}|$. Referring to Case 1 in Appendix A.4, we have the following utility disparity upper bound for AI confidence levels $\{h_2, a_2\}$:*

$$\begin{aligned}
0 &\le \left|\mathbb{E}_\pi\left[u(T,Y)|a_2, z \in \mathcal{Z}_{h_2,1}\right] - \mathbb{E}_\pi\left[u(T,Y)|a_2, z \in \mathcal{Z}_{h_2,0}\right]\right| \\
&\le \left(1 - \frac{\alpha}{2}\right) \cdot (u(1,0) - u(1,1) - u(0,0) + u(0,1)) \cdot \Delta_1 \\
&\quad + \frac{\alpha}{2} \cdot (u(1,1) - u(1,0) - u(0,1) + u(0,0)).
\end{aligned} \tag{91}$$

*where,*

$$\Delta_1 = P(Y = 1|a_1, z \in \mathcal{Z}'_{h_1,1}) - P(Y = 1|a_1, z \in \mathcal{Z}'_{h_1,0})$$
$$- P(Y = 1|a_2, z \in \mathcal{Z}'_{h_2,1}) + P(Y = 1|a_2, z \in \mathcal{Z}'_{h_2,0}).$$

*Using the conditions given by Eqs. (89) and (90), we have:*

$$-2\alpha_g \le \Delta_2 \le 2\alpha_g. \tag{92}$$

*where,*

$$\Delta_2 = P(Y = 1|a_1, z \in \mathcal{Z}'_{h_1,1} \cap \mathcal{Z}''_{h_1,1}) - P(Y = 1|a_1, z \in \mathcal{Z}'_{h_1,0} \cap \mathcal{Z}''_{h_1,0})$$
$$- P(Y = 1|a_2, z \in \mathcal{Z}'_{h_2,1} \cap \mathcal{Z}''_{h_2,1}) + P(Y = 1|a_2, z \in \mathcal{Z}'_{h_2,0} \cap \mathcal{Z}''_{h_2,0}).$$

*For $z \in \mathcal{Z}'_h \setminus \mathcal{Z}''_h$, it holds that,*

$$-2 \le \Delta_3 \le 2. \tag{93}$$

*where,*

$$\Delta_3 = P(Y = 1|a_1, z \in \mathcal{Z}'_{h_1,1} \setminus \mathcal{Z}''_{h_1,1}) - P(Y = 1|a_1, z \in \mathcal{Z}'_{h_1,0} \setminus \mathcal{Z}''_{h_1,0})$$
$$- P(Y = 1|a_2, z \in \mathcal{Z}'_{h_2,1} \setminus \mathcal{Z}''_{h_2,1}) + P(Y = 1|a_2, z \in \mathcal{Z}'_{h_2,0} \setminus \mathcal{Z}''_{h_2,0}).$$

*Incorporating these conditions into the utility disparity upper bound in Eq. (91), we get:*

$$0 \le |\mathbb{E}_\pi [u(T,Y)|a_2, z \in \mathcal{Z}_{h_2,1}] - \mathbb{E}_\pi [u(T,Y)|a_2, z \in \mathcal{Z}_{h_2,0}]|$$
$$\le (u(1,1) - u(1,0) - u(0,1) + u(0,0)) \cdot \left( \left(1 - \frac{\alpha_h}{2}\right) \cdot \left( \left(1 - \frac{\alpha_g}{2}\right) \cdot 2\alpha_g + \frac{\alpha_g}{2} \cdot 2 \right) + \frac{\alpha_h}{2} \right)$$
$$= (u(1,1) - u(0,1) - u(1,0) + u(0,0)) \cdot \left( \frac{\alpha_h}{2} + \left(1 - \frac{\alpha_h}{2}\right) \cdot \left(3\alpha_g - \alpha_g^2\right) \right). \tag{94}$$

*This bound can be similarly derived for Cases 2~4 in Appendix A.4, yielding a consistent utility disparity bound as stated in Theorem 3.6.* $\square$

## A.6 PROOF OF THEOREM 4.5

**Theorem 4.5** Suppose $f_A(z)$ is $\alpha/2$-cognition-aware multicalibrated with respect to the collection $\mathcal{C} = \{\mathcal{Z}_{h,s_i}\}_{h \in \mathcal{H}, s_i \in \mathcal{S}}$. Then, $f_A$ satisfy both $\alpha$-human-alignment and $\alpha$-inter-group-alignment.

***Proof.*** *According to the definition of cognition-aware multicalibration in Method 4.3, when human decision-makers are partitioned into two groups with group membership values $s = 0$ and $s = 1$, and given a fixed human confidence level $h \in \mathcal{H}$, there exists a subset $\mathcal{Z}'_h \subset \mathcal{Z}_h$ with $|\mathcal{Z}'_h| \ge \left(1 - \frac{\alpha}{2}\right) \cdot |\mathcal{Z}_h|$ such that, for any AI confidence $a_1, h_1 \in [0, 1]$, it holds that,*

$$\left| P(Y = 1 \mid f_A(z) = a_1, z \in \mathcal{Z}'_{h_1,1}) - a_1 \right| \le \frac{\alpha}{2}. \tag{95}$$

$$\left| P(Y = 1 \mid f_A(z) = a_1, z \in \mathcal{Z}'_{h_1,0}) - a_1 \right| \le \frac{\alpha}{2}. \tag{96}$$

*From the given inequalities, we have:*

$$P(Y = 1 \mid f_A(z) = a_1, z \in \mathcal{Z}'_{h_1,1}) \in \left[a_1 - \frac{\alpha}{2}, a_1 + \frac{\alpha}{2}\right], \tag{97}$$

$$P(Y = 1 \mid f_A(z) = a_1, z \in \mathcal{Z}'_{h_1,0}) \in \left[a_1 - \frac{\alpha}{2}, a_1 + \frac{\alpha}{2}\right]. \tag{98}$$

*Then, it's natural that $f_A$ satisfied $\alpha$-inter-group-alignment as,*

$$\left| P(Y = 1 \mid f_A(z) = a_1, z \in \mathcal{Z}'_{h_1,1}) - P(Y = 1 \mid f_A(z) = a_1, z \in \mathcal{Z}'_{h_1,0}) \right| \le \alpha. \tag{99}$$

*As for any human decision-maker confidence $0 \le h_1 \le h_2 \le 1$ and $0 \le a_1 \le a_2 \le 1$, it holds that,*

$$P(Y = 1 \mid f_A(z) = a_2, z \in \mathcal{Z}'_{h_2,1}) \in \left[a_2 - \frac{\alpha}{2}, a_2 + \frac{\alpha}{2}\right]. \tag{100}$$

$$P(Y = 1 \mid f_A(z) = a_2, z \in \mathcal{Z}'_{h_2,0}) \in \left[a_2 - \frac{\alpha}{2}, a_2 + \frac{\alpha}{2}\right]. \tag{101}$$

*For $P(S = 1|a, z \in \mathcal{Z}'_h) + P(S = 0|a, z \in \mathcal{Z}'_h) = 1$, it holds that,*

$$
\begin{aligned}
& P(Y = 1|a_1, z \in \mathcal{Z}'_{h_1}) - P(Y = 1|a_2, z \in \mathcal{Z}'_{h_2}) \\
& = P(Y = 1|a_1, z \in \mathcal{Z}'_{h_1,1}) \cdot P(S = 1|a_1, z \in \mathcal{Z}'_{h_1}) \\
& + P(Y = 1|a_1, z \in \mathcal{Z}'_{h_1,0}) \cdot P(S = 0|a_1, z \in \mathcal{Z}'_{h_1}) \\
& - P(Y = 1|a_2, z \in \mathcal{Z}'_{h_2,1}) \cdot P(S = 1|a_2, z \in \mathcal{Z}'_{h_2}) \\
& - P(Y = 1|a_2, z \in \mathcal{Z}'_{h_2,0}) \cdot P(S = 0|a_2, z \in \mathcal{Z}'_{h_2}) \\
& = P(Y = 1|a_1, z \in \mathcal{Z}'_{h_1,1}) \cdot P(S = 1|a_1, z \in \mathcal{Z}'_{h_1}) \\
& + P(Y = 1|a_1, z \in \mathcal{Z}'_{h_1,0}) \cdot \left(1 - P(S = 1|a_1, z \in \mathcal{Z}'_{h_1})\right) \\
& - P(Y = 1|a_2, z \in \mathcal{Z}'_{h_2,1}) \cdot P(S = 1|a_2, z \in \mathcal{Z}'_{h_2}) \\
& - P(Y = 1|a_2, z \in \mathcal{Z}'_{h_2,0}) \cdot \left(1 - P(S = 1|a_2, z \in \mathcal{Z}'_{h_2})\right) \\
& \leq P(S = 1|a_1, z \in \mathcal{Z}'_{h_1}) \cdot (a_1 + \frac{\alpha}{2}) + \left(1 - P(S = 1|a_1, z \in \mathcal{Z}'_{h_1})\right) \cdot (a_1 + \frac{\alpha}{2}) \\
& - P(S = 1|a_2, z \in \mathcal{Z}'_{h_2}) \cdot (a_2 - \frac{2}{\alpha}) - \left(1 - P(S = 1|a_2, z \in \mathcal{Z}'_{h_2})\right) \cdot (a_2 - \frac{2}{\alpha}) \\
& = \alpha + a_1 - a_2 \leq \alpha.
\end{aligned}
\tag{102}
$$

*Consequently, $f_A$ satisfies $\alpha$-human-alignment and $\alpha$-inter-group-alignment meanwhile.* □

## A.7 EXPERIMENT SETTINGS

### A.7.1 DATASET PROCESSING

Following the data processing Corvelo Benz & Rodriguez (2023), we transform the original dataset's confidence values from a scale of $[-1, 1]$ to $[0, 1]$ to ensure consistency with our human-AI inter-active model (Section 2). In the original dataset, predictions by participants are from different but overlapping sets of countries across tasks, who are told the AI advice has different accuracy. Thus, to control for these confounding factors, we focus exclusively on participants from the United States who are informed that the AI advice had an $80\%$ accuracy. Furthermore, we use education level as a sensitive attribute, a recommended factor that influence cognitive capacities of human decision-makers across different tasks but should be treated with equal utility in AI assistance for social good Zappalà et al. (2024); Ward et al. (2022).

### A.7.2 DISCRETIZATION

Following the consistent experimental settings outlined in Corvelo Benz & Rodriguez (2023), we provide a detailed description of the discretization process used in our experiments: the human con-fidence $h$ is discretized into 3 bins per task, $\{\mathcal{H}_1, \mathcal{H}_2, \mathcal{H}_3\}$, corresponding to low, medium, and high confidence levels, respectively. The bin boundaries are set such that each bin contains ap-proximately equal probability mass, with the bin values assigned as the average confidence within each bin. The AI's confidence $a$ are divided into uniformly sized bins per task with centred value given by $\Lambda = \left\{\frac{\lambda}{2}, \frac{3\lambda}{2}, \dots, 1 - \frac{\lambda}{2}\right\}$, where $\lambda = 1/8$. The above process discretizes the continu-ous confidence space $\mathcal{H} \times \mathcal{A}$ into a grid of $3 \times \lfloor 1/\lambda \rfloor$ cells. The $(i, j)$-th grid cell contains the samples $\mathcal{Z}^j_i = \{(x, h, s) \in \mathcal{Z} | h \in \mathcal{H}_i, f_A(z) \in [\Lambda[j] - \lambda/2, \Lambda[j] + \lambda/2]\}$. Furthermore, based on the value of the sensitive attribute $s \in \mathcal{S}$, the samples in $\mathcal{Z}^j_i$ can be further divided into $\mathcal{Z}^j_i = \left\{\mathcal{Z}^j_{i,1}, \mathcal{Z}^j_{i,0}\right\}$, where $\mathcal{Z}^j_{i,1} = \{(x, h, s) \in \mathcal{Z} | h \in \mathcal{H}_i, f_A(z) \in [\Lambda[j] - \lambda/2, \Lambda[j] + \lambda/2), S = 1\}$ and $\mathcal{Z}^j_{i,0} = \{(x, h, s) \in \mathcal{Z} | h \in \mathcal{H}_i, f_A(z) \in [\Lambda[j] - \lambda/2, \Lambda[j] + \lambda/2), S = 0\}$.

## A.8 ADDITIONAL EXPERIMENTS

### A.8.1 ADDITIONAL QUANTITATIVE ABLATION RESULTS

Tables 6 and 7 provide detailed quantitative results corresponding to the visualizations in Figures 1 and 2. The key takeaways are:

Table 6: Decision Utility: Ablation quantification results (Black: mean; Blue: standard deviation).

| Exp. | No Adjust | | | Method 4.4 | | | Method 4.3 | | |
|---|---|---|---|---|---|---|---|---|---|
| | AI-only | human-only | AI-assisted | AI-only | human-only | AI-assisted | AI-only | human-only | AI-assisted |
| 1 | 0.7843 | 0.6601 | 0.7570 | 0.8163 | 0.6601 | 0.8068 | 0.8221 | 0.6601 | 0.8045 |
| | (1.3e-15) | (0) | (1.1e-16) | (1.3e-15) | (0) | (0.006) | (7.7e-16) | (0) | (0.005) |
| 2 | 0.8082 | 0.7366 | 0.7755 | 0.8163 | 0.6601 | 0.8068 | 0.8586 | 0.7366 | 0.8253 |
| | (5.5e-16) | (3.3e-16) | (5.5e-16) | (1.6e-15) | (3.3e-16) | (0.004) | (3.3e-16) | (3.3e-16) | (0.005) |
| 3 | 0.7596 | 0.7293 | 0.7792 | 0.8008 | 0.7293 | 0.7973 | 0.8089 | 0.7293 | 0.8025 |
| | (3.3e-16) | (0) | (1.5e-15) | (1.1e-16) | (0) | (0.003) | (5.5e-16) | (0) | (0.003) |
| 4 | 0.7596 | 0.7293 | 0.7792 | 0.8052 | 0.7103 | 0.7845 | 0.7967 | 0.7103 | 0.7817 |
| | (1.1e-16) | (1.2e-15) | (3.3e-16) | (6.6e-16) | (1.2e-15) | (0.004) | (8.8e-16) | (1.2e-15) | (0.004) |

Table 7: Decision Utility Disparity: Ablation quantification results (Black: mean; Blue: standard deviation).

| Exp. | No Adjust | | | Method 4.4 | | | Method 4.3 | | |
|---|---|---|---|---|---|---|---|---|---|
| | AI-only | human-only | AI-assisted | AI-only | human-only | AI-assisted | AI-only | human-only | AI-assisted |
| 1 | 0.0426 | 0.0441 | 0.0420 | 0.0362 | 0.0441 | 0.0467 | 0.0304 | 0.0441 | 0.0382 |
| | (9.0e-17) | (5.5e-17) | (9.0e-17) | (0) | (5.5e-17) | (0.0038) | (6.2e-17) | (5.5e-17) | (0.0081) |
| 2 | -0.0468 | -0.0507 | -0.0595 | -0.0226 | -0.0507 | -0.0148 | 0.0185 | -0.0507 | 0.0014 |
| | (8.3e-17) | (2.7e-17) | (9.0e-17) | (3.4e-18) | (2.7e-17) | (0.0037) | (2.4e-17) | (2.7e-17) | (0.0072) |
| 3 | -0.0374 | 0.0410 | 0.0203 | -0.0347 | 0.0410 | -0.0075 | -0.0136 | 0.0410 | -0.0047 |
| | (3.4e-17) | (4.1e-17) | (1.0e-17) | (4.1e-17) | (4.1e-17) | (0.0024) | (5.2e-18) | (4.1e-17) | (0.0056) |
| 4 | 0.0049 | 0.0164 | 0.0222 | 0.0077 | 0.0164 | 0.0314 | -0.0352 | 0.0164 | 0.0089 |
| | (4.3e-18) | (1.7e-17) | (1.7e-17) | (1.1e-17) | (1.7e-17) | (0.0058) | (2.7e-17) | (1.7e-17) | (0.0060) |

**Decision utility improvement:** Both *Cognition-unaware Multicalibration* (Method 4.4) and *Cognition-aware Multicalibration* (Method 4.3) enhance AI-only decision utility through AI confidence adjustment. Given that human-only decision utility remains fixed, this improvement in AI-only utility translates to higher overall AI-assisted decision performance, providing more substantial assistance to human decision-makers across all groups.

**Decision utility disparity reduction:** Compared to both the *No Adjust* and *Cognition-unaware Multicalibration*, *Cognition-aware Multicalibration* effectively *reduces or reverses* decision utility disparities across different human decision-maker groups in AI-only settings, ultimately achieving substantial fairness improvements in decision utility under AI assistance.

### A.8.2 GROUP-LEVEL ACCURACY DISTRIBUTION ANALYSIS

Figure 4 shows the instances' decision accuracy distributions across different decision-maker groups. The results demonstrate that *Cognition-aware Multicalibration* achieves fairness by delivering better instance outcomes for the disadvantaged group (Group $S=1$ with lower "education" level) without negatively impacting the advantaged group (Group $S=0$).

### A.8.3 IMPACT OF DISCRETIZATION PARAMETERS

Figure 5 presents the evaluation of EAE and EIAE metrics across $4$ tasks after performing cognition-aware multicalibration with varying hyperparameters $\widetilde{\alpha} \sim [0.0001, 0.01, 0.1]$ and $\lambda \sim [0.1, 0.125, 0.2]$. The results indicate a general trend where smaller values of $\widetilde{\alpha} + \lambda$ (reflected in results closer to the left or bottom of the plots) lead to lower EAE and EIAE values. We focus on the general trend as we have claimed that discrete evaluation metrics like EAE and EIAE provide a more accurate reflection of alignment when there are significant changes and may fail to capture alignment across the entire continuous confidence space fully. Nevertheless, the general trend strongly supports the notion that decreasing $\widetilde{\alpha} + \lambda$ leads to improved *human-alignment* and *inter-group-alignment*.

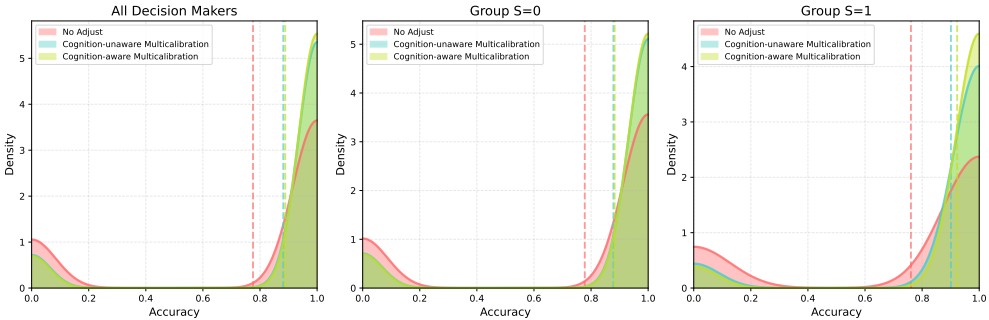

Figure 4: Instance accuracy distributions across all decision makers, Group $S=0$, and Group $S=1$ (lower "education" level) on the Cities task.

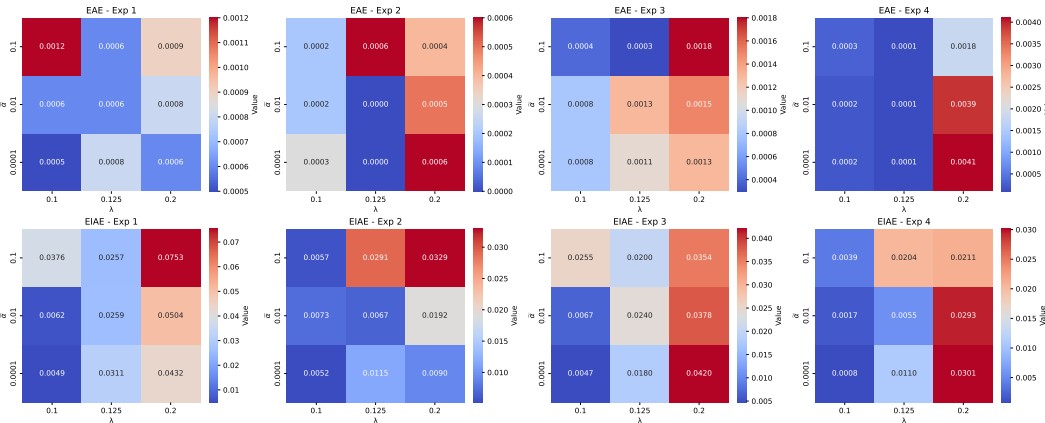

Figure 5: Visualization of EAE and EIAE metrics following cognition-aware multicalibration, with parameters $\widetilde{\alpha} \sim [0.0001, 0.01, 0.1]$ and $\lambda \sim [0.1, 0.125, 0.2]$.

### A.8.4 GENERALIZATION TO MULTIPLE GROUPS

In this experiment, we introduce "gender" as an additional demographic feature for grouping ("Female" as Group $S = 0$ and "Male" as Group $S = 1$). We first validate the effectiveness of our method under the new demographic feature "gender". The experimental results are presented in Figures 6 and 7. We further conduct experiments considering both "gender" and "education" as demographic features, resulting in 4 human decision-maker groups: $S = (0,0)$, $S = (0,1)$, $S = (1,0)$, and $S = (1,1)$. The utility disparity is measured using the standard deviation of utility distributions across these subgroups:

$$\text{Disp} = \text{Std}(\mathbb{E}[\mathbf{1}(T = Y) \mid S = (0,0)], \mathbb{E}[\mathbf{1}(T = Y) \mid S = (0,1)], \\ \mathbb{E}[\mathbf{1}(T = Y) \mid S = (1,0)], \mathbb{E}[\mathbf{1}(T = Y) \mid S = (1,1)]). \tag{103}$$

The results are presented in Figures 8 and 9.

Across these experiments, where human decision-makers are grouped by either single or multiple demographic features, the results align with expectations and demonstrate the following key takeaways:

1. Effectiveness in improving fairness across different settings: Across both single-group and multi-group settings, *Cognition-aware Multicalibration* consistently outperforms both *No adjust* and *Cognition-unaware Multicalibration* in terms of fairness. Notably, it avoids the fairness deterioration observed in *Cognition-unaware Multicalibration* (i.e., tasks 1, 3 and 4 in Figure 6, task 3 in Figure 8).

2. Comparable utility performance: *Cognition-aware Multicalibration* achieves utility performance comparable to *Cognition-unaware Multicalibration*. For example, in Figure 6,

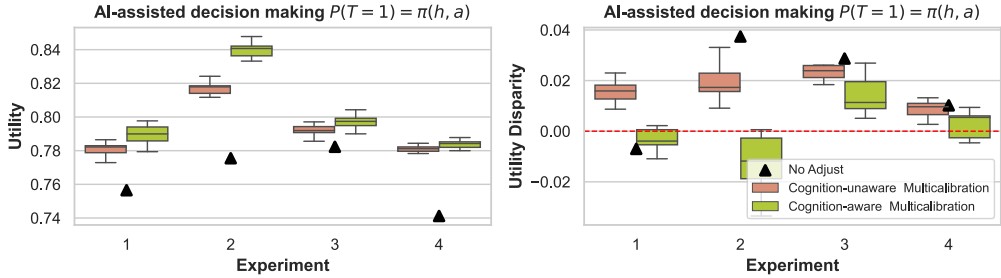

Figure 6: Grouped by "gender": Statistics of utility and utility disparity over 100 experiments, where the final decision $P(T = 1) = \pi(h, a)$ is made by human with AI assistance. The AI confidence is adjusted by *No Adjust*, *Cognition-unaware Multicalibration* and *Cognition-aware Multicalibration*, respectively.

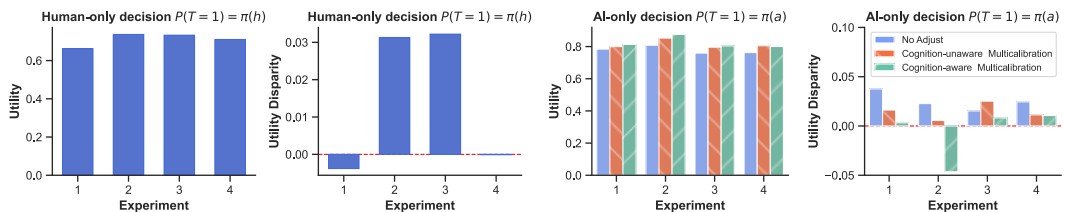

Figure 7: Grouped by "gender": The utility and utility disparity where the final decision $P(T = 1) = \pi(h)$ is made by human-only or AI-only, respectively.

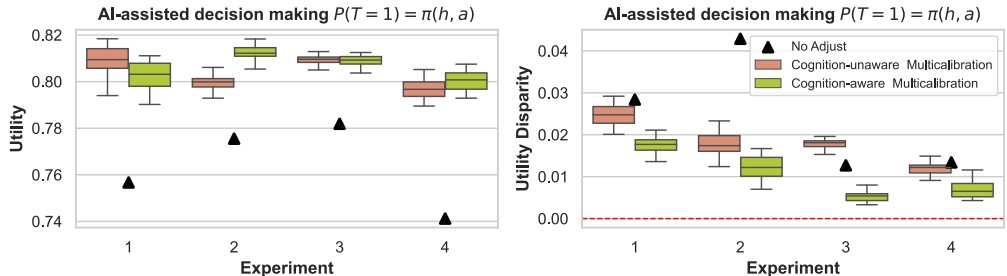

Figure 8: Grouped by both "gender" and "education": Statistics of utility and utility disparity over 100 experiments, where the final decision $P(T = 1) = \pi(h, a)$ is made by human with AI assistance. The AI confidence is adjusted by *No Adjust*, *Cognition-unaware Multicalibration* and *Cognition-aware Multicalibration*, respectively.

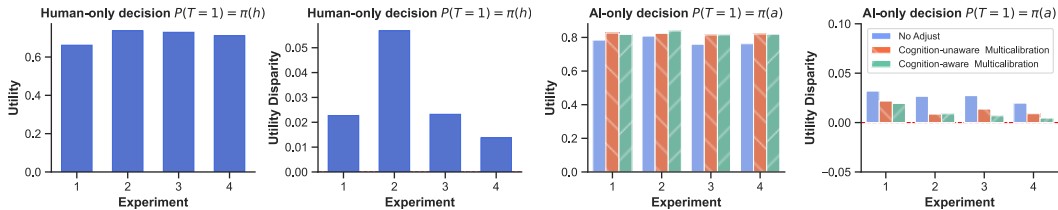

Figure 9: Grouped by both "gender" and "education": The utility and utility disparity where the final decision $P(T = 1) = \pi(h)$ is made by human-only or AI-only, respectively.

*Cognition-aware Multicalibration* demonstrates similar utility performance compared to *Cognition-unaware Multicalibration*. In tasks 2, and 4 of Figures 8, *Cognition-aware Multicalibration* shows superior utility performance. Although there is a slight accuracy drop in task 1, with the 25th percentile decreasing by $0.9\%$ compared to *Cognition-unaware Mul-*

Table 8: Runtime comparison (in seconds).

| Exp. | Method 4.4 | Method 4.3+2 subgroups | | | Method 4.3+4 subgroups | | | | |
|---|---|---|---|---|---|---|---|---|---|
| | Overall | Overall | Group 0 | Group 1 | Overall | Group 0 | Group 1 | Group 2 | Group 3 |
| 1 | 1.397 | 1.714 | 0.682 | 1.032 | 1.050 | 0.272 | 0.427 | 0.207 | 0.144 |
| 2 | 1.372 | 1.480 | 1.065 | 0.415 | 1.131 | 0.273 | 0.295 | 0.300 | 0.262 |
| 3 | 1.092 | 1.246 | 0.958 | 0.288 | 0.714 | 0.036 | 0.195 | 0.329 | 0.152 |
| 4 | 0.502 | 1.062 | 0.921 | 0.140 | 0.806 | 0.087 | 0.094 | 0.299 | 0.324 |

*ticalibration*, this trade-off is considered acceptable given the significant gains in fairness across groups.

3. Adjusting AI confidence to mitigate utility disparities: *Cognition-aware Multicalibration* effectively adjusts AI confidence to reduce or reverse utility disparities in AI-only decisions, compensating for disparities observed in human-only decisions. This behavior remains consistent across different group settings (Figures 7 and 9).

### A.8.5 COMPUTATIONAL COMPLEXITY ANALYSIS

For each sensitive group $s \in \mathcal{S}$, the Method 4.3 iteratively updates AI confidence values within each grid cell $(i, j, s)$ until convergence, where $i \in \{1, \ldots, N_A\}$ denotes the index of AI bins, $j \in \{1, \ldots, N_H\}$ denotes the index of human bins, $N_A = \lfloor 1/\lambda \rfloor$ denotes the number of AI confidence bins, $N_H$ denotes the number of human confidence bins, and $n_{i,j,s}$ denotes the number of instances in cell $(i, j, s)$. For each cell $(i, j, s)$, the Method 4.3 update the AI confidence for $n_{i,j,s}$ instances until the tolerance threshold $\widetilde{\alpha}$ is met. By the convergence guarantee of $\lambda$-discretization (Hebert-Johnson et al., 2018), the algorithm terminates in at most $O(1/(\widetilde{\alpha} \cdot \lambda))$ rounds, where each round processes all $n_{i,j,s}$. Therefore, the computation complexity of the computation in each cell $(i, j, s)$ is $O(n_{i,j,s}/(\widetilde{\alpha} \cdot \lambda))$. The total time complexity over all cells is $O(N/(\widetilde{\alpha} \cdot \lambda))$, as $\sum_{i,j,s} n_{i,j,s} = n_{all}$, where $n_{all}$ is the total number of instances. Specifically in Table 8, we compare the runtime of Method 4.4 with Method 4.3 using 2 subgroups and 4 subgroups, respectively. The experimental results reveal no significant correlation between overall runtime and the number of subgroups. Instead, runtime variation is primarily driven by differences in per-group computation time. This variation arises mainly from group-specific data distributions that affect the initial miscalibration and result in different convergence times within each subgroup.

### A.8.6 MULTI-CLASS CLASSIFICATION

Our current theoretical results focus on binary classification, as binary decision tasks encompass a wide range of AI-assisted decision-making scenarios, including loan approval, medical diagnosis, and hiring decisions. In this part, we discuss potential ways to extend our framework for providing fair and high-utility AI-assisted decision-making in multi-class settings.

For the multi-classification utility function defined in Equation (104), we employ One-Vs-All (OVA) decomposition to reduce the multi-class problem to $K$ binary subproblems. The proofs presented in our work are modular and can be directly applied.

$$u(T, Y) = \begin{cases} r^+ & \text{if } T = Y \\ r^- & \text{if } T \neq Y, \end{cases} \tag{104}$$

where $r^- < r+$. For a K-class classification problem with labels $Y^{\text{all}} \in \{0, 1, ..., K - 1\}$, we decompose it into $K$ mutually exclusive binary subproblems: (1) "*Is the label 0?*", ..., (K) "*Is the label K − 1?*". Each binary subproblem k is associated with a binary label $Y^{(k)} \in \{0, 1\}$, where $Y^{(k)} = \mathbb{I}(Y^{\text{all}} = k)$. The labels satisfy mutual exclusivity: for any instance $i$, exactly one binary label is positive across all subproblems, $\sum_{k=0}^{K-1} Y^{(k)} = 1$, thus, $Y^{(k)} = 1 \rightarrow Y^{\text{all}} = k$, we have,

$$P(Y^{(k)} = 1 | h^{(k)}, a^{(k)}) = P(Y^{\text{all}} = k | h^{(0)}, a^{(0)}, ..., h^{(K-1)}, a^{(K-1)}). \tag{105}$$

For each binary subproblem, the decision policy is denoted as $P(T^{(k)} = 1 | h^{(k)}, a^{(k)}) = \pi^{(k)}(h^{(k)}, a^{(k)})$. When the AI confidence is calibrated to perfect human-alignment such that

Table 9: Performance evaluation of **multi-class decision-making** on the Art task (Black: mean; Blue: standard deviation).

| Classes | No adjust | | Method 4.4 | | Method 4.3 | |
|---|---|---|---|---|---|---|
| | Utility | Utility Disparity | Utility | Utility Disparity | Utility | Utility Disparity |
| 3 | 0.749 | 0.049 | 0.758 | 0.052 | 0.775 | 0.027 |
| | (0.010) | (0.005) | (0.006) | (0.005) | (0.005) | (0.005) |
| 5 | 0.460 | 0.035 | 0.495 | 0.035 | 0.505 | 0.026 |
| | (0.004) | (0.002) | (0.002) | (0.005) | (0.003) | (0.003) |

$\pi^{(k)}(h^{(k)}, a^{(k)})$ is monotonically related to $P(Y^{(k)} = 1|h^{(k)}, a^{(k)})$, the multi-class optimal decision is:

$$T^* = \arg\max_k \pi^{(k)}(h^{(k)}, a^{(k)}) = \arg\max_k P(Y_{\text{all}} = k|h^{(0)}, a^{(0)}, ..., h^{(K-1)}, a^{(K-1)}). \quad (106)$$

Equation (106) is Bayes optimal: it selects the class with the highest posterior probability. Consequently, **optimal utility in each binary subproblem guarantees optimal multi-class utility (Rifkin & Klautau, 2004) when the *argmax* rule is applied with perfect human-alignment calibrated confidences**.

Similarly, when each binary subproblem achieves optimal fairness, that is, minimize utility disparity $\text{UD}_k \to 0$ via perfect inter-group-alignment, the utility disparity in multi-classification problem is bounded by:

$$|\mathbb{E}[u(T, Y_{all})|s = 1] - \mathbb{E}[u(T, Y_{all})|s = 0]|$$
$$\leq \sum_{k=0}^{K-1} P(T^* = k) \cdot \left|\mathbb{E}[u(T, Y^{all})|T^* = k, s = 1] - \mathbb{E}[u(T, Y^{all})|T^* = k, s = 0]\right|. \quad (107)$$

Since $Y^{(k)} = \mathbb{I}(Y^{\text{all}} = k)$ and $u(T, Y^{\text{all}})|T^* = k$ represents the utility when class $k$ is selected (not necessarily correctly classified), it holds that: $\mathbb{E}[u(T, Y^{all})|T^* = k] = \mathbb{E}[u(T^{(k)}, Y^{(k)})]$. Therefore, Equation (107) can be further bounded as:

$$|\mathbb{E}[u(T, Y_{all})|s = 1] - \mathbb{E}[u(T, Y_{all})|s = 0]| \leq \sum_{k=0}^{K-1} P(T^* = k) \cdot UD_k, \quad (108)$$

where $UD_k$ denotes the utility disparity when class $k$ is selected:

$$UD_k = \left|\mathbb{E}[u(T^{(k)}, Y^{(k)})|s = 1] - \mathbb{E}[u(T^{(k)}, Y^{(k)})|s = 0]\right|. \quad (109)$$

The above process establishes that **fairness in each binary subproblem guarantees fairness in the multi-class setting for decision-makers with heterogeneous cognitive capacities**.

Table 9 presents a performance comparison of three methods for 3-class and 5-class classification on the Art task. The experimental results demonstrate that *Cognition-aware Multicalibration* effectively improves both overall utility and utility fairness across decision-maker groups.

### A.9 SCOPE AND FUTURE WORK

#### A.9.1 FAIRNESS METRICS

In this work, we use utility disparity for measuring fair utility, aligning with the decision-makers' primary goal of making more accurate decisions with AI assistance (Steyvers & Kumar, 2024). The concept of utility disparity is also evident in centralized learning (Chi et al., 2021) and decentralized learning (Donahue & Kleinberg, 2023). While other fairness metrics, such as demographic parity and equalized odds (Mehrabi et al., 2021), emphasize the equality of positive or true positive outcomes, these metrics primarily relate to individuals being judged (e.g., patients) and diverge from the fair utility objective for decision-makers (e.g., doctors). Nonetheless, exploring diverse fairness concepts remains an interesting avenue for future work, particularly in understanding the underlying motivations of fairness principles beyond utility disparity in the context of AI-assisted decision-making.

### A.9.2 Archival Dataset Usage

Our evaluation employs archival human-AI interaction datasets, a methodological choice grounded in cognitive science showing that humans maintain consistent decision-making strategies when addressing similar problems or within short-term interactions (Goldstein, 2015; Cacciabue et al., 1992), and following established precedent in the human-AI collaboration area, where archival datasets serve as the standard for algorithmic validation (Vodrahalli et al., 2022b; Corvelo Benz & Rodriguez, 2023).

As the first work addressing utility fairness for human decision-makers with cognitive disparities, the archival-dataset approach is sufficient for establishing and validating our core theoretical and methodological contributions: ① identifying utility unfairness in AI-assisted decision-making among cognitively heterogeneous human decision-makers, ② developing cognition-aware multi-calibration grounded in the theoretically-founded objectives of inter-group-alignment and human-alignment, and ③ demonstrating substantial improvements in decision utility fairness compared to baselines while maintaining comparable overall utility.

While live user studies would introduce confounding factors arising from subjective human behavior—challenges that exceed the scope of our fairness-focused investigation, we view carefully controlled live studies as a valuable future direction to investigate the impact of dynamic human behavioral adaptation on AI-assisted decision-making performance, building upon the theoretical and algorithmic foundations established in this work.

### A.9.3 Connection to Game-Theoretic Perspectives

In this section, we discuss the relationship between our proposed framework and game-theoretic solution concepts, their applicability differences, and potential for integration.

**Stackelberg equilibrium and strategic interaction.** Our work addresses fairness issue in AI-assisted decision-making by adjusting the AI confidence towards theoretically-founded alignment objectives. This represents an *AI-side intervention* approach. In contrast, the Stackelberg game model characterizes AI-assisted decision-making as a sequential strategic interaction where the AI system (leader) first commits to a policy, and human decision-makers (followers) subsequently respond by optimizing their own strategies given the AI's policy. Under this game-theoretic perspective, the focus shifts to *strategically influencing human behavior*. These represent two fundamentally different problem-solving paradigms, distinguished by their assumptions about human adaptability. Our work establishes a solid theoretical foundation for utility fairness in AI-assisted decision-making, grounded in humans employing consistent decision-making strategies. This is supported by cognitive science showing that humans typically maintain consistent strategies when addressing similar problems or within short-term interactions (Goldstein, 2015; Cacciabue et al., 1992). Game-theoretic approaches assume humans can and will strategically adapt their behavior in response to AI policies. Modeling such strategic adaptation and the resulting equilibrium dynamics presents additional complexities that exceed the scope of our fairness-focused investigation. The appropriate choice between these different problem-solving routes depends on the application context and whether decision-makers are willing and able to modify their strategies.

While our current work focuses on stable human behavior (Goldstein, 2015; Cacciabue et al., 1992), we note that longer-term interactions involving repeated AI assistance may exhibit dynamic behavioral evolution. Investigating such scenarios where humans strategically adapt over time represents a valuable future research direction. A game-theoretic perspective could be beneficially integrated with our proposed AI-side alignment framework to account for dynamic changes in human decision-making behavior.

**Shapley value.** The Shapley value is a solution concept in cooperative game theory that allocates the total payoff generated by a coalition fairly among its members in proportion to their marginal contributions. While the Shapley value has been successfully applied to fairness problems in machine learning contexts, it does not naturally apply to our AI-assisted decision-making setting due to fundamental structural differences. AI-assisted decision-making involves no collective payoff requiring allocation. Instead, each decision-maker operates independently on their own instances, deriving utility exclusively from their individual decisions. Moreover, the decision utility is inher-

ently non-transferable, and the performance of one decision-maker cannot be redistributed to benefit or harm others.

### A.9.4 Yule's Effect in Subgroups

As highlighted by Ruggieri et al. (2023), achieving fairness at the group level does not necessarily guarantee fairness at finer granularities—a phenomenon known as the *Yule's effect*. This phenomenon arises because, beyond the directly observable grouping attributes, such as education level we adopted, there may exist additional confounding variables that contribute to within-group heterogeneity. As it is difficult to determine whether a feature is a confounder or a mediator, Kamiran et al. (2013) proposed clustering strata into a manageable number of groups to control for confounding factors. Such subgroup identification and stratification methods are orthogonal to our work and can be naturally integrated: our framework provides the calibration objectives for AI confidence and determines how AI confidence is adjusted within groups, while stratification methods refine the human decision-maker grouping by accounting for additional confounding variables. As demonstrated in our experiments (Appendix A.8.4), when we partition users into multiple subgroups, *Cognition-aware Multicalibration* continues to effectively reduce decision utility disparities within each subgroup. This makes the combination feasible and promising for future work, to ensure that fairness improvements do not cause utility drops for specific sub-populations within the broader cognitive capacity groups we define.

