# OpenReview forum: "Fair Decision Utility in Human-AI Collaboration: Interpretable Confidence Adjustment for Humans with Cognitive Disparities"
_ICLR.cc/2026/Conference — ICLR 2026 Poster_

### Official Review · Reviewer_MdV1 · 2025-10-14

**Soundness:** 3
**Presentation:** 3
**Contribution:** 3
**Rating:** 6
**Confidence:** 4

**Summary:**

In this paper, the authors explore how humans with different characteristics may derive unfair utility from the same advice. They introduce the new concept of inter-group alignment and propose an interpretable, fairness-aware objective for AI confidence adjustment. Their analysis shows that achieving utility fairness in AI-assisted decision making requires both human alignment and inter-group alignment. Building on these objectives, they present a multicalibration-based AI confidence adjustment approach tailored to scenarios involving decision-makers with heterogeneous cognitive capacities. Evaluations on four datasets with real human-behavior data demonstrate that the proposed method significantly improves utility fairness across groups without sacrificing overall utility.

**Strengths:**

1. The paper is clear, well organised, and adds several fresh ideas to human-AI interaction research. It first defines human-alignment and the new inter-group alignment, proves why both matter, and then shows an elegant multicalibration fix to improve the fairness utility across the group.

2. The experiment design and evaluation is thorough to justify the main claim of this paper.

3. The authors publicly release their code, making replication and future extensions straightforward for other researchers.

**Weaknesses:**

1. The framework assumes people can report confidence on a common, well-calibrated scale, yet prior work shows self-reported confidence is often noisy and inconsistent across individuals.

2. All experiments rely on archival human-AI datasets; the paper does not test how live users react to the adjusted confidences or how their perceptions and strategies might change.

3. The evaluation covers only binary decision tasks. Extending the theory and method to multi-class settings would broaden its practical reach.

**Questions:**

See above weakness.

---

> ### Author Response · Authors · 2025-11-21
> **Response to Reviewer MdV1 [1]**
>
> We thank the reviewer for the thoughtful feedback and recognizing our clear presentation, fresh ideas, thorough experiments, and public code release for reproducibility. We address each of your comments point-by-point below.
>
> ### W1: [...] self-reported confidence is often noisy and inconsistent
>
> We appreciate this important question. We would like to clarify that our framework does not assume people report confidence on a common, well-calibrated scale. In fact, **the inconsistency of self-reported confidence across individuals is not a limitation but precisely the  problem we have addressed**. Our framework explicitly models this through heterogeneous cognitive capacities, which captures how different groups exhibit systematically different confidence reporting behaviors. Our alignment objectives  are specifically designed to adjust AI confidence to account for these group-specific differences, thereby achieving fair utility across diverse groups despite their inconsistent interpretations and uses of self-reported confidence.
>
>
> Moreover, **our empirical validation uses real human-reported confidence collected from actual decision-making tasks, which inherently contains both  inconsistencies and random noise**. The consistent improvements in utility fairness  across all four tasks demonstrate that our method is robust to the variability in real-world confidence reporting.
>
>
> ### W2: All experiments rely on archival human-AI datasets [...]
>
> We appreciate this comment and `have added a detailed discussion on archival dataset usage in Appendix A.9.2`. We would like to clarify several important points:
>
> (1) Our methodological choice of archival datasets is grounded in both theoretical justification and established practice. **Cognitive science research demonstrates that humans maintain consistent decision-making strategies when addressing similar problems or within short-term interactions [1,2], which supports the validity of archival datasets for evaluating AI assistance interventions**. Moreover, **our approach follows established precedent in the human-AI collaboration literature, where archival datasets serve as the standard methodology for algorithmic validation [3,4]**.
>
> (2) **As the first work addressing utility fairness for human decision-makers with cognitive disparities, the archival-dataset approach is sufficient and appropriate for establishing and validating our core contributions**: (1) identifying utility unfairness arising from cognitively heterogeneous human decision-makers, (2) developing cognition-aware multicalibration grounded in theoretically-founded alignment objectives, and (3) demonstrating substantial improvements in decision utility fairness while maintaining comparable overall utility. These foundational theoretical and algorithmic contributions can be rigorously validated through archival datasets containing real human decisions.
>
> (3) While live user studies would be valuable for investigating dynamic behavioral adaptation over extended interactions, they would introduce substantial confounding factors arising from subjective human behavior—**challenges that exceed the scope of our fairness-focused investigation**. **We view carefully controlled live studies as an important future direction to build upon the foundations established in this work.**
>
>
> [1] E Bruce Goldstein. Cognitive psychology: Connecting mind, research, and everyday experience. Cengage learning Stamford, CT, 2015.
>
> [2] P.C. Cacciabue, F. Decortis, B. Drozdowicz, M. Masson, and J.-P. Nordvik. Cosimo: A cognitive simulation model of human decision making and behavior in accident management of complex plants. IEEE Transactions on Systems, Man, and Cybernetics, 22(5):1058–1074, 1992.
>
> [3] Kailas Vodrahalli, Tobias Gerstenberg, and James Y Zou. Uncalibrated models can improve human-ai collaboration. In Advances in Neural Information Processing Systems, volume 35, pp. 4004–4016. Curran Associates, Inc., 2022b.
>
> [4] Nina Corvelo Benz and Manuel Rodriguez. Human-aligned calibration for ai-assisted decision making. In A. Oh, T. Naumann, A. Globerson, K. Saenko, M. Hardt, and S. Levine (eds.), Advances in Neural Information Processing Systems, volume 36, pp. 14609–14636. Curran Associates, Inc., 2023.

---

> > ### Author Response · Authors · 2025-11-21
> > **Response to Reviewer MdV1 [2]**
> >
> > ### W3: The evaluation covers only binary decision tasks [...]
> >
> > We appreciate this valuable suggestion. `We have added a comprehensive extension to multi-class settings in Appendix A.8.6.` We would like to clarify several important points:
> >
> >
> > (1) For the multi-class utility function (i.e., $u(T,Y) = r^+$ if $T=Y$, $r^-$ if $T\neq Y$ and $r^-< r^+$), **our theories and methods can be extended to multi-class classification**.
> > >We  employ One-Vs-All (OVA) decomposition to reduce the multi-class problem to $K$ binary subproblems.  The proofs presented in our work are modular and can be directly applied. For a $K$-class problem, we decompose it into $K$ binary subproblems, each asking "Is the label $k$?" with $k=0,...,K-1$. Our theoretical results modularly apply to each binary subproblem. Critically, we prove that (a) when AI confidence achieves perfect human-alignment in each binary subproblem, the decision rule $T^* = \arg max_k \pi^{(k)}(h^{(k)}, a^{(k)})$ is Bayes optimal, guaranteeing that optimal utility in the binary subproblems leads to optimal utility in multi-class classification; and (b) when each binary subproblem achieves optimal fairness via inter-group-alignment $UD_k \rightarrow0$, the multi-class utility disparity is bounded by $\sum_k P(T^*=k)·UD_k \rightarrow 0$, thus fairness in each binary subproblem guarantees fairness in the multi-class setting.
> >
> > (2) **We have provided `Table 9 in Appendix A.8.6`, which presents our method's performance validation under 3-class and 5-class classification on the Art task**. The results demonstrate that Cognition-aware Multicalibration effectively improves both overall utility and utility fairness in multi-class settings. For example, in 3-class classification, our method improves utility from 0.758 (maximum of baselines) to 0.775 while reducing utility disparity from 0.049 (minimum of baselines) to 0.027. **These results validate that our framework successfully extends beyond binary tasks.**

---

> > > ### Comment · Reviewer_MdV1 · 2025-11-21
> > >
> > > Thanks for your response! I will maintain my score.

---

### Official Review · Reviewer_T784 · 2025-10-16

**Soundness:** 4
**Presentation:** 4
**Contribution:** 3
**Rating:** 8
**Confidence:** 3

**Summary:**

The paper tackles fairness in human–AI collaboration from the angle of ensuring that AI assistance provides equitable benefits to human decision-makers who possess heterogeneous cognitive capacities. The authors argue that existing alignment notions such as calibration and "human-alignment" are insufficient for guaranteeing fairness *across* decision-maker groups. They introduce the concept of "inter-group-alignment" to capture disparities in decision utility between subpopulations of humans. The paper provides a formal theoretical analysis linking human alignment and inter-group alignment. Building on this insight, the authors propose "cognition-aware multicalibration", and prove that it serves as a sufficient condition for simultaneously achieving both alignment objectives.
The claims are validated through experiments on four human-AI decision-making tasks. The results demonstrate that the proposed method successfully reduces utility disparities across groups while maintaining overall performance.

**Strengths:**

1. **Novel and Interesting Perspective:** The paper shifts fairness focus from data subjects to the decision-makers themselves. It's an underexplored and insightful perspective as it illuminates how AI systems can inadvertently widen performance gaps between varied decision-makers (e.g., experts and novices).
2. **Elegant Theoretical Framework with Principled Operationalization:** The derivation of a utility disparity bound provides an interpretable bridge between fairness notions, offering practitioners a clear diagnostic for assessing group-level disparities. Cognition-aware multicalibration operationalizes the theory in a computationally practical way.
3. **Empirical Credibility:** Validation on multiple real-world tasks demonstrates both robustness and practical feasibility. Additional experiments in appendix demonstrate thoughtfulness and rigor.

**Weaknesses:**

1. **Scalability Challenges with Multi-Attribute Grouping:** The proposed framework requires partitioning the data and calibrating for each subgroup. While the experiments show this is feasible for a small number of groups, the computational and data requirements would grow exponentially with the number of sensitive attributes. A deeper discussion of the practical limits (e.g., the number of subgroups that can be realistically handled) would be beneficial.

2. **Rational Decision-Making and Static Setting:** The theoretical claims hinge on the assumption that humans follow a rational, monotone decision policy (Assumption 2.1). However, a large body of literature shows that human behavior often deviates from pure rationality, exhibiting various cognitive biases. The framework does not currently account for such suboptimal but more realistic decision policies. The analysis assumes that the cognitive capacities and decision policies of the human groups are static. In many real-world collaborations, humans learn and adapt their strategies based on repeated AI interactions. The current framework does not address these dynamic or online contexts, where calibration needs might evolve over time.

**Questions:**

I found the paper to be insightful and well-executed. The weaknesses section already incorporate most questions, primarily aimed at better understanding the practical implications and boundaries of the proposed framework. One additional question pertains to incorporating broader notions of utility within proposed framework. The paper defines utility as decision accuracy. In human-AI collaboration, however, overall utility can also encompass factors like decision time, cognitive load, or the user's trust and self-efficacy. How do you see the concepts of human-alignment and inter-group-alignment extending to these more multi-faceted, human-centric notions of utility?

---

> ### Author Response · Authors · 2025-11-21
> **Response to Reviewer T784 [1]**
>
> We thank the reviewer for the thoughtful feedback and for recognizing the novel decision-maker-centered fairness perspective, the elegant and interpretable theoretical framework, and the robust empirical validation. We address each of your comments point-by-point below.
>
> ### W1: Scalability Challenges with Multi-Attribute Grouping
>
> We appreciate your question and `have added a detailed Computational Complexity Analysis in the Appendix A.8.5 to address this concern comprehensively.` We would like to clarify the key points:
>
> (1) Theoretical Complexity: Our algorithm processes instances in each grid cell $(i,j,s)$ until convergence, and the total computation across all cells sums to $O(N/(\widetilde \alpha \cdot \lambda))$ because $\sum_{i,j,s} n_{i,j,s} =n_{all}$, where $n_{i,j,s}$ denotes the number of instances in cell $(i,j,s)$, $n_{all}$ is the total number of instances, $\widetilde \alpha $ is the tolerance threshold, and $\lambda$ is the discretization parameter. Critically, **this complexity is independent of the number of subgroups**;
>
> (2) Empirical Validation: `Table 8 in Appendix A.8.5` compares runtime for 2 and 4 subgroups across our 4 tasks. The results show **no exponential growth, or even significant correlation, between runtime and the number of subgroups**.  The results empirically demonstrate that our method scales efficiently with the number of subgroups in practice.
>
> ### W2: Rational Decision-Making and Static Setting
>
> We thank you for this thoughtful comment. `We provide a detailed discussion of the rationale and scope of Assumption 2.1 in Appendix A.1`.  Importantly, our empirical findings reveal that without any explicit instruction to follow rational or monotonic policies, **human decision-makers in real-world tasks naturally exhibit this monotonic behavior**: monotonicity violation rates across our $4$ tasks are only **0.20\%-0.53\%** (`Table 3 in Appendix A.1`). This demonstrates that our assumption is **descriptive of natural human behavior** rather than an unrealistic requirement. Even under artificially added noise that makes  monotonicity violation rates of $1.9\%-4.9\%$ in `Appendix A.1-Table 4`, our **Cognition-aware Multicalibration maintains competitive utility while significantly improving fairness** as shown in `Appendix A.1-Figure 3` and `Appendix A.1-Table 5`, showing **robustness to realistic deviations**.
>
>  Regarding the static setting, **the setting is well-grounded in cognitive science showing humans maintain stable decision policies within similar tasks or short timeframes [1,2]**. This applies to many important real-world scenarios, i.e., helping to make timely diagnoses within a short term.`We acknowledge in Appendix A.9.3 that longer-term scenarios involving strategic adaptation represent important future work, where game-theoretic perspectives could be integrated with our alignment framework.` However, **the static setting addresses a substantial and practically important portion of current AI-assisted decision-making deployments, and our work provides the first systematic treatment of fairness in this context**.
>
> [1] E Bruce Goldstein. Cognitive psychology: Connecting mind, research, and everyday experience. Cengage learning Stamford, CT, 2015.
>
> [2] P.C. Cacciabue, F. Decortis, B. Drozdowicz, M. Masson, and J.-P. Nordvik. Cosimo: A cognitive simulation model of human decision making and behavior in accident management of complex plants. IEEE Transactions on Systems, Man, and Cybernetics, 22(5):1058–1074, 1992.

---

> > ### Author Response · Authors · 2025-11-21
> > **Response to Reviewer T784 [2]**
> >
> > ### Question: How do you see the concepts of human-alignment and inter-group-alignment extending to these more multi-faceted, human-centric notions of utility? [...]  like decision time, cognitive load, or the user's trust and self-efficacy [...]
> >
> > We thank you for this excellent question.
> >
> > **We would like to clarify that our framework currently focuses specifically on decision accuracy as utility, as it represents the fundamental objective in  AI-assisted decision-making tasks**: making correct predictions. This focus is important and well-justified for several reasons. First, decision outcome utility directly reflects the core value proposition of AI assistance—improving the correctness of human decisions in high-stakes domains like medical diagnosis, credit assessment, and legal judgments. Second, our theoretical framework fundamentally relies on the relationship between confidence and decision outcomes—this natural connection enables our alignment objectives to be  practically implementable through confidence adjustment.
> >
> > Regarding trust,  **fairness is a critical factor influencing user trust in AI systems** [1]. In this sense, our work on improving utility fairness can positively contribute to building trust.
> >
> > Regarding the other dimensions you mentioned, i.e., decision time, cognitive load, self-efficacy, are indeed important aspects of human-AI interaction, but **they operate through different mechanisms that extend beyond confidence calibration**. For example, cognitive load has been explored through approaches such as interface design [2] and information presentation format [3]. While confidence presentation may have indirect effects on these dimensions, they are not primarily governed by confidence adjustments. **We view this as valuable future work that could build upon our framework toward building more comprehensive AI-assisted decision-making systems that address both `the dimensions you mentioned and the utility fairness dimension we focus on in this work`**.
> >
> > Importantly, **achieving  decision utility fairness is a foundational and first step** to benefit users and encourage their willingness to engage with and adopt AI-assisted systems.
> >
> > [1] Sacharidis, Dimitris. "Fairness and explainability for enabling trust in ai systems." A Human-Centered Perspective of Intelligent Personalized Environments and Systems. Cham: Springer Nature Switzerland, 2024. 85-110.
> >
> > [2] Faudzi, Masyura Ahmad, et al. "User interface design in mobile learning applications: Developing and evaluating a questionnaire for measuring learners' extraneous cognitive load." Heliyon 10.18 (2024).
> >
> > [3] Yin, Jiaqi, Tiong-Thye Goh, and Yi Hu. "Using a chatbot to provide formative feedback: A longitudinal study of intrinsic motivation, cognitive load, and learning performance." IEEE Transactions on Learning Technologies 17 (2024): 1378-1389.

---

### Official Review · Reviewer_Wqv4 · 2025-10-30

**Soundness:** 3
**Presentation:** 2
**Contribution:** 2
**Rating:** 6
**Confidence:** 2

**Summary:**

The paper demonstrates that existing AI confidence adjustment objectives, such as calibration and human alignment, may lead to utility disparities across groups of decision-makers with varying cognitive capacities.  Moreover, the paper shows that the utility disparity between human decision-maker groups is bounded by a function of human-alignment level and inter-group-alignment level. Hence, the authors propose  a multicalibration-based AI confidence adjustment approach to mitigate this concern, validating their approach over four different datasets.

**Strengths:**

The main strengths are:

1. The paper covers an overlooked problem in the human-AI alignment literature, as it focuses on fairness in human-AI collaboration
2. The proposed multicalibration approach is principled and based on the proposed theory
3. Overall, the paper's goal is well-motivated and interesting

**Weaknesses:**

The main shortcomings of the paper are:

*Significance* While I appreciate the theoretical results, I think this new paradigm comes with some implementation choices that are not straightforward. For instance, in real-life scenarios, evaluating cognitive disparities might be even unethical, so such information might never be disclosed. Moreover, it is also quite challenging to assess if improving the fairness for decision-maker groups leads to better outcomes for the observed instances; hence, I think the authors should discuss these possible limitations.

*Assumptions Discussion* I think the authors should discuss Assumption 2.1 better, as understanding when such an assumption might fail is a key factor for the proposed method.

*Empirical Evaluation:* I think some of the results are not very clear. For instance, I would suggest that the authors reformat Table 1, as it does not report standard deviations and incorrectly displays the best entries in bold.

**Questions:**

I would like the authors to discuss my highlighted shortcomings.

Moreover, I wonder if the authors could detail some settings where $P(Y|S=1)$ is different from $P(Y|S=0)$? My intuition is that, for some reason, the instances that are considered by one group can be sampled from different parts of the feature space, but I want to be sure about it.

---

> ### Author Response · Authors · 2025-11-21
> **Response to Reviewer Wqv4 [1]**
>
> We sincerely thank the reviewer for the thoughtful feedback and for recognizing that our work addresses an overlooked  problem, proposes a principled approach, and is well-motivated. We address each of your comments point-by-point below.
>
> ### Comment:  [...] Evaluating cognitive disparities might be even unethical  [...]
>
> We sincerely thank you for raising this important implement consideration. We would like to clarify that in AI-assisted decision-making scenarios, understanding human capabilities is often `essential` for achieving effective human-AI complementarity. We believe that `evaluating cognitive disparities can be conducted ethically with proper safeguards`, including:  (1) Users must be informed about how their interaction data is analyzed; (2) Users should have the option to opt-in or opt-out of capability assessment; (3) AI systems must explicitly state that cognitive capability evaluation is used solely to provide improved and fairer AI assistance, not for discriminatory purposes; (4) All capability assessments should be kept confidential.
>
> Moreover, our framework operate at the **group level**, further reduces privacy and ethical concerns while still enabling substantial fairness improvements.
>
> ### Comment: [...] challenging to assess if improving the fairness for decision-maker groups leads to better outcomes for the observed instances [...]
>
> We thank you for raising this question. To clearly address it, `we have added instance decision accuracy distribution analysis in Appendix A.8.2`. The results demonstrate that Cognition-aware Multicalibration achieves fairness by delivering better instance outcomes for the disadvantaged group (Group S=1 with lower education) without negatively impacting the advantaged group (Group S=0). **The results provide clear evidence that improving decision utility fairness in AI-assisted decision making does not make one group's instances suffer to benefit others**.
>
>
> ### Comment: Assumptions Discussion [...]
>
> We thank you for this valuable suggestion. Following your suggestion, `we have added a detailed discussion on the rationale and scope of Assumption 2.1 in Appendix A.1`.
>
> Regarding the validity of the assumption: Assumption 2.1 describes the behavior of rational decision-makers who  aim to maximize decision utility: they increase their likelihood of making a positive decision $T=1$ when both their own confidence $h$ and AI confidence $a$ increase.  While Assumption 2.1 could be violated in atypical scenarios—such as AI aversion, where decision-makers systematically decrease their decision likelihood as AI confidence increases, or random errors where decision-makers accidentally make incorrect decisions—such cases represent departures from rational utility-maximizing behavior and are outside our current scope.
>
> **Importantly, this assumption holds widely in real-world settings.** Our empirical analysis across all four real-world AI-assisted decision-making tasks reveals that `Assumption 2.1 holds remarkably well in practice`. We computed pairwise violation rates by checking whether $P(T_i=1) > P(T_j=1)$ when $h_i \leq h_j$ and $a_i \leq a_j$. As shown in `Appendix A.1-Table 3`, violation rates are consistently low: 0.20%-0.53% across all tasks, providing `strong empirical evidence that rational monotonic behavior naturally emerges in real-world human-AI collaboration`.
>
> **To further validate the robustness of the proposed method to the violation of assumptions,** we constructed instances set with artificially elevated violation rates 1.9\%-4.9\% as in `Appendix A.1-Table 4`—approximately 4-10× higher than natural occurrence. Even under these intentionally noisy conditions, our **Cognition-aware Multicalibration maintains competitive utility while significantly improving fairness as in `Appendix A.1-Figure 3` and `Appendix A.1-Table 5`**. This demonstrates that our framework **remains effective even when monotonicity violations increase substantially beyond naturally observed levels, addressing practical concerns about occasional deviations from the assumption in deployed systems**.
>
> ### Comment: [...] reformat Table 1, as it does not report standard deviations and incorrectly displays the best entries in bold.
>
> We thank you for this important suggestion. Following your suggestion, **we have added standard deviations to `Table 1`**. The standard deviations are very small and can be considered negligible. Additionally, **we have  highlighted the best entries in `Table 1`**. **The key takeaway remains unchanged**: the proposed Cognition-aware Multicalibration achieves competitive human-alignment while attaining optimal inter-group-alignment.

---

> > ### Author Response · Authors · 2025-11-21
> > **Response to Reviewer Wqv4 [2]**
> >
> > ### Question: [...] detail some settings where $P(Y|S=1)$ is different from $P(Y|S=0)$ ? [...]
> >
> > Thank you for your question. We would like to clarify that our paper does not involve P(Y|S=1) being different from P(Y|S=0). We believe you are referring to $P(Y=1|S=1) \neq P(Y=1|S=0)$ in  Lemma A.1 and Lemma A.2 , which **corresponds to the scenario you described**.  In practice, `different human decision-makers, especially when grouped by cognitive capabilities, may observe instances with different features`.   For example, highly experienced doctors are more often assigned instances with more severe symptoms, leading to different positive rates compared to others.

---

> ### Comment · Reviewer_Wqv4 · 2025-11-23
>
> I thank the authors for their response, most of my concerns have been addressed. After reading the whole author-reviewer discussion, I have a few final remarks:
>
> > The results demonstrate that Cognition-aware Multicalibration achieves fairness by delivering better instance outcomes for the disadvantaged group (Group S=1 with lower education) without negatively impacting the advantaged group (Group S=0). **The results provide clear evidence that improving decision utility fairness in AI-assisted decision making does not make one group's instances suffer to benefit others.**
>
> I would be more cautious regarding these results. As highlighted in Ruggieri et al., 2023, statistical paradoxes such as the Yule effect or the Simpson paradox can occur when conditioning on subgroups: even if the aggregate utility for the advantaged group remains stable, the intervention might cause significant utility drops for specific sub-populations within that group (or within the disadvantaged group) that are canceled out when averaging.
> I think a similar line of reasoning might apply here, so I think it might be worth highlighting possible Yule effects as a potential limitation of the current work.
>
> Second, since the authors present several metrics (some where lower is better, others where higher is better), I would encourage adding the direction of the effect directly in the Table captions
>
>
> I am willing to increase my score once these two final remarks are taken into account.
>
>
> Ruggieri, S., Alvarez, J. M., Pugnana, A., State, L., & Turini, F. (2023, February). Can we trust fair-AI?. In Proceedings of the Thirty-Seventh AAAI Conference on Artificial Intelligence.

---

> ### Author Response · Authors · 2025-11-23
> **Response to Reviewer Wqv4**
>
> We appreciate your constructive suggestions and are pleased that most of your concerns have been addressed.  Following your suggestions, we have made corresponding revisions.
>
> > ### Regarding the Yule Effect:
>
> For the grouping attribute `education level` explicitly adopted in our experiments, our results in `Appendix A.8.2` demonstrate that cognition-aware multicalibration achieves fairness improvements without negatively impacting the advantaged group `at each group's aggregate level`. We acknowledge the reviewer's comment about potential statistical paradoxes. While developing metrics or grouping methods to account for finer-grained confounding variables is not the primary focus of our work, we recognize it as an important consideration for fairness research.
>
> **Following your suggestion, we have added a detailed discussion in Appendix A.9.4**, where we:
>
> (1) highlight the potential existence of Yule's effect;
>
> (2) clarify a promising direction for integrating our cognition-aware multicalibration framework with existing stratification method. Our framework provides the calibration objectives for AI confidence  and determines how AI confidence is adjusted within groups, while stratification methods refine the human decision-maker grouping by accounting for additional confounding variables.
>
>
> > ### Regarding Table Captions:
>
> We have revised Table 1 and Figure 1 captions to explicitly indicate the direction of improvement for each metric.
>
> Please see our updated manuscript for these changes.

---

> > ### Comment · Reviewer_Wqv4 · 2025-11-27
> >
> > I thank the authors for their further clarification. Hence, I am increasing the score to a full accept.

---

> > > ### Author Response · Authors · 2025-11-27
> > >
> > > Thank you for taking the time to review our work and for the score increase.

---

### Official Review · Reviewer_wqhf · 2025-10-31

**Soundness:** 4
**Presentation:** 3
**Contribution:** 4
**Rating:** 8
**Confidence:** 5

**Summary:**

This paper studies fairness in AI-assisted decision-making, focusing on situations where humans have different cognitive capacities. The authors show that existing approaches, such as calibration and human-alignment, do not necessarily ensure utility fairness across different groups of human decision-makers.

They introduce a new concept called inter-group-alignment, which ensures that human groups with similar confidence levels receive comparable decision utility when supported by the AI. The paper provides solid theoretical analysis, deriving a clear upper bound on utility disparity as a function of both human-alignment and inter-group-alignment.

To achieve this dual goal in practice, they propose a Cognition-aware Multicalibration algorithm that adjusts AI confidences accordingly. Experiments on four real-world datasets (Art, Cities, Sarcasm, and Census) confirm that this method significantly reduces fairness gaps between human groups while maintaining overall performance.

The theoretical formulation, clarity of objectives, and validation through experiments make this a valuable contribution to the fairness and human-AI collaboration literature.

**Strengths:**

The introduction of inter-group-alignment provides a fresh and rigorous way to think about fairness in human-AI collaboration.

The paper is theoretically complete, with well-defined assumptions and proofs that connect intuitively to fairness.

The proposed Cognition-aware Multicalibration method is interpretable, practical, and mathematically justified.

The empirical validation is strong: four datasets, clear fairness and utility metrics, and consistent results showing improvements.

The work bridges human and algorithmic fairness, showing how cognitive differences among people can be formally addressed rather than ignored.

**Weaknesses:**

Monotonicity Assumption (Assumption 2.1):
This assumption states that humans act rationally, increasing their probability of making a positive decision as either their own or the AI’s confidence increases. While this is reasonable for theoretical proofs, it may not always hold in practice.
Real humans are often non-monotonic in their decision behavior. For instance, they may overtrust or undertrust AI advice due to biases, cognitive fatigue, or misunderstanding of confidence. In such cases, increasing the AI’s confidence may not increase the chance of a positive decision, which violates monotonicity.

The authors could strengthen the paper by (a) discussing how this assumption might break in real-world conditions, (b) analyzing the theoretical impact if monotonicity is only approximately satisfied, and (c) outlining how the proposed method could still remain effective. Even a small simulation with “noisy” or biased human decision policies would illustrate robustness and make the work more realistic and influential.

Ablation Study Clarity:
The paper does include human-only, AI-only, and human-AI results, but these are only mentioned in Figure 2 and not summarized in a table. It would be much clearer to show numerical comparisons between these three conditions across datasets. This would highlight how the collaboration truly benefits fairness and overall utility.

Hyperparameter Sensitivity:
The appendix covers sensitivity analyses for parameters like $\tilde{\alpha}$ and $\lambda$, but the main text should summarize them briefly. Readers need to know how robust the fairness improvements are if these parameters change. A short paragraph or figure in the main paper would make the method easier to trust and reproduce.

Game-Theoretic Perspective:
The problem naturally relates to concepts like Stackelberg games (AI as leader, human as follower) or Shapley-value-based fairness allocation. A short paragraph connecting the proposed framework to these ideas would situate the work more clearly within the broader literature on incentive alignment and cooperative fairness.

Behavioral Impact of AI Adjustments:
Since the AI modifies its confidence outputs to improve fairness, it would be interesting to consider whether this adjustment might influence how humans behave over time. For example, could one group become overly reliant on AI due to consistently higher confidence signals? Even a brief reflection on this would enhance the paper’s real-world applicability.

**Questions:**

Could you provide a clear quantitative table comparing human-only, AI-only, and human-AI decisions across all datasets to make the ablation more explicit?

How sensitive are the results to the parameters $\tilde{\alpha}$ and $\lambda$? Would adaptive tuning or data-driven adjustment improve generalization?

If the human decision policy is not strictly monotonic (for example, due to inconsistent trust or cognitive biases), how does that affect your theoretical bounds or empirical fairness outcomes?

Could the dual-alignment framework extend naturally to multi-class or regression tasks?

Have you considered a Stackelberg or cooperative-game interpretation of your framework, where the AI strategically adjusts its outputs to optimize a fairness-aware social welfare objective? Are they feasible to perform and related at all?

Do you think modifying AI confidence distributions could unintentionally change how different human groups rely on the AI (either over-trusting or disengaging)?

---

> ### Author Response · Authors · 2025-11-21
> **Response to Reviewer wqhf [1]**
>
> We sincerely thank the reviewer for the thoughtful feedback and for recognizing the fresh perspective, theoretical rigor, and practical contributions of our work. We address each of your comments point-by-point below.
>
> ### Comment: Monotonicity Assumption (Assumption 2.1)
>
> #### (a) [...] discussing how this assumption might break in real-world conditions
>
> We appreciate this question and `have added a detailed discussion on the rationale and scope of Assumption 2.1 in Appendix A.1`. We discussed that the monotonicity assumption could theoretically be violated in atypical scenarios,  such as AI aversion, where decision-makers systematically decrease their decision likelihood as AI confidence increases, or random errors where decision-makers accidentally make incorrect decisions. However, such cases represent departures from rational utility-maximizing behavior. **Critically, our empirical analysis across all four real-world AI-assisted decision-making tasks reveals that Assumption 2.1 holds remarkably well in practice.** We computed pairwise violation rates by checking whether $P(T_i=1) > P(T_j=1)$ when $h_i \leq h_j$ and $a_i \leq a_j$. As shown in `Appendix A.1 Table 3`, violation rates are consistently low: 0.20%-0.53% across all tasks, providing  **empirical evidence that rational monotonic behavior naturally emerges in real-world AI-assisted decision-making**.
>
> #### (b) [...] analyzing the theoretical impact if monotonicity is only approximately satisfied?
>
> We thank you for this insightful suggestion. We would like to provide a theoretical analysis of the impact when monotonicity is only approximately satisfied below:
> > Consider two confidence levels $(a,h)$ and $(a',h')$ where $a < a'$ and $h < h'$. Under perfect human-alignment, we have $P(Y=1|a,h) = p_y < p_y' = P(Y=1|a',h')$. Under the monotonicity assumption, $P(T=1|a',h') = \pi' ≥ \pi = P(T=1|a,h)$, and the decision utility at $(a',h')$ is: $U_{\text{mono}} = \pi'[p_{y}' \cdot u(1,1) + (1-p_{y}') \cdot u(1,0)] + (1-\pi')[p_{y}'\cdot u(0,1) + (1-p_{y}') \cdot u(0,0)]$.When monotonicity is approximately satisfied with violation degree $\epsilon$, such that, $P(T=1|a',h') = \pi' - \epsilon < \pi = P(T=1|a,h)$, the decision utility becomes: $U_{\text{violation}} = (\pi' - \epsilon)[p_{y}' \cdot u(1,1) + (1-p_{y}') \cdot u(1,0)] + (1-\pi'+\epsilon)[p_{y}' \cdot u(0,1) + (1-p_{y}') \cdot u(0,0)]$. The utility loss is:$\Delta U = U_{\text{mono}} - U_{\text{violation}} = \epsilon \cdot \{p_{y}'[u(1,1) - u(0,1)] + (1-p_{y}')[u(1,0) - u(0,0)]\}$. Given the utility function properties  $u(1,1) > u(0,1)$ and $u(1,0) < u(0,0)$, as $p_{y}'$ increases (which occurs at higher confidence levels under human-alignment), $\Delta U$ approaches positive values. This demonstrates that violations of monotonicity lead to utility loss, with the magnitude determined by  $\epsilon$ and task-specific utility values $u(1,1), u(0,1), u(1,0), u(0,0)$, thereby affecting the achievable optimal utility. However, our empirical analysis above shows that in real-world AI-assisted decision-making tasks, $\epsilon$ is extremely small.
>
> #### (c) [...] “noisy” or biased human decision policies would illustrate robustness [...]
>
> To  validate robustness, **we constructed instance sets with artificially elevated monotonicity assumption violation rates of 1.9\%-4.9\%, as shown in `Appendix A.1 Table 4`—approximately 4-10$\times$ higher than natural occurrence**. Even under these intentionally noisy conditions, our Cognition-aware Multicalibration maintains competitive utility while significantly improving fairness as in `Appendix A.1-Figure 3` and `Appendix A.1-Table 5`. This demonstrates that our framework **remains effective even when monotonicity violations increase substantially beyond naturally observed levels, addressing practical concerns about occasional deviations from the assumption**.
>
>
> ### Comment: [...] The appendix covers sensitivity analyses for parameters, but the main text should summarize them briefly [...]
>
> We thank you for this suggestion. `We have modified this in Section 5.1-Hyperparameters`, where we summarize the findings from parameter sensitivity analysis: smaller $\widetilde{\alpha} + \lambda$ values lead to improved human-alignment and inter-group-alignment.
>
> ### Question: [...] quantitative table comparing human-only, AI-only, and human-AI decisions across all datasets to make the ablation more explicit [...]
>
> We thank you for this valuable suggestion. `We have added detailed quantitative ablation results in Appendix A.8.1 to make the ablation more explicit`, and **the results are consistent with our previous findings**: Compared to both the No Adjust baseline and Cognition-unaware Multicalibration, Cognition-aware Multicalibration effectively reduces or reverses decision utility disparities across different human decision-maker groups in AI-only settings, ultimately achieving substantial fairness improvements.

---

> > ### Author Response · Authors · 2025-11-21
> > **Response to Reviewer wqhf [2]**
> >
> > ### Question: How sensitive are the results to the parameters ? Would adaptive tuning or data-driven adjustment improve generalization?
> >
> > Thank you for this question. `Appendix A.8.3` presents comprehensive sensitivity analysis across  $\widetilde{\alpha} \in [0.0001, 0.01, 0.1]$ and $\lambda \in [0.1, 0.125, 0.2]$, showing that smaller $\widetilde{\alpha} + \lambda$  improve both human-alignment and inter-group-alignment. We agree that adaptive tuning could potentially be beneficial. However, given the clear monotonic relationship between $\widetilde{\alpha} + \lambda$  and alignment performance, and the low-dimensional parameter space (only two parameters), reducing $\widetilde{\alpha} + \lambda$ provides an effective and practical guideline for parameter selection. This simple strategy eliminates the need for complex adaptive tuning procedures while achieving strong empirical performance across all our experiments.
> >
> > ### Question: [...] human decision policy is not strictly monotonic, how does that affect your theoretical bounds or empirical fairness outcomes?
> >
> > We appreciate this question. As discussed comprehensively in our response to `Comment: Monotonicity Assumption (Assumption 2.1)` above, Our framework exhibits  robustness to violations of monotonicity assumption—even under artificially elevated violation rates significantly higher than naturally observed levels, our method maintains competitive utility while achieving  fairness improvements.
> >
> >
> > ### Question: [...] extend naturally to multi-class or regression tasks?
> >
> > We thank you for this insightful question.
> >
> > `We have added a comprehensive extension to multi-class settings in Appendix A.8.6`. We would like to clarify several important points:
> > (1) **Our dual-alignment framework can be theoretically extended to multi-class settings.**
> > > For the multi-class utility function (i.e., $u(T,Y) = r^+$ if $T=Y$, $r^-$ if $T\neq Y$ and $r^-< r^+$), **our theories and methods can be extended to multi-class classification**. We  employ One-Vs-All (OVA) decomposition to reduce the multi-class problem to $K$ binary subproblems.  The proofs presented in our work are modular and can be directly applied.
> >  For a $K$-class problem, we decompose it into $K$ binary subproblems, each asking "Is the label $k$?" with $k=0,...,K-1$. Our theoretical results modularly apply to each binary subproblem. Critically, we prove that (1) when AI confidence achieves perfect human-alignment in each binary subproblem, the decision rule $T^* = \arg max_k \pi^{(k)}(h^{(k)}, a^{(k)})$ is Bayes optimal, guaranteeing that optimal utility in the binary subproblems leads to optimal utility in multi-class classification; and (2) when each binary subproblem achieves optimal fairness via inter-group-alignment $UD_k \rightarrow0$, the multi-class utility disparity is bounded by $\sum_k P(T^*=k)·UD_k \rightarrow 0$, thus fairness in each binary subproblem guarantees fairness in the multi-class setting.
> >
> > (2) We have provided `Appendix A.8.6-Table 9`, which **presents our method's performance validation under 3-class and 5-class classification on the Art task.**
> > > The results demonstrate that Cognition-aware Multicalibration effectively improves both overall utility and utility fairness in multi-class settings. For example, in 3-class classification, our method improves utility from 0.758 (maximum of baselines) to 0.775 while reducing utility disparity from 0.049 (minimum of baselines) to 0.027. These results validate that our framework successfully extends beyond binary tasks.
> > >
> > (3) For **regression tasks**, additional challenges arise: continuous decision spaces make discrete utility functions less natural, and AI-assisted regression scenarios are uncommon in practice—most applications use direct AI predictions without human decision-making involvement. We view these extensions as important future work.

---

> ### Author Response · Authors · 2025-11-21
> **Response to Reviewer wqhf [3]**
>
> ### Comment & Question: Game-Theoretic Perspective [...] Have you considered a Stackelberg or cooperative-game interpretation?
>
> We thank you for this valuable suggestion. `We have added a detailed discussion on the connection to game-theoretic perspectives in Appendix A.9.3`. We would like to clarify several important points:
>
>  (1) Relationship to Stackelberg equilibrium and strategic interaction. Our work addresses fairness issue in AI- assisted decision-making by adjusting the AI  confidence towards theoretically-founded alignment objectives. This represents an **AI-side intervention** approach. In contrast, the Stackelberg game model characterizes AI-assisted decision-making as a sequential strategic interaction where the AI system (leader) first commits to a policy, and human decision-makers (followers) subsequently respond by optimizing their own strategies given the AI's policy. Under this game-theoretic perspective, the focus shifts to strategically **influencing human behavior**. **These represent two fundamentally different  problem-solving routes, distinguished by their assumptions about human adaptability**. Our work establishes a solid theoretical foundation for utility fairness in AI-assisted decision-making, grounded in humans employing consistent decision-making strategies. This is supported by **cognitive science showing that humans typically maintain consistent strategies when addressing similar problems or within short-term interactions [1,2]**. Game-theoretic approaches assume humans can and will strategically adapt their behavior in response to AI policies. Modeling such strategic adaptation and the resulting equilibrium dynamics presents additional complexities that exceed the scope of our fairness-focused investigation. **The appropriate choice between these different problem-solving routes depends on the application context and whether decision-makers are willing and able to modify their strategies.** While our current work focuses on  stable human behavior [1,2], we note that longer-term interactions involving repeated AI assistance may exhibit dynamic behavioral evolution. Investigating such scenarios where humans strategically adapt over time represents a valuable future research direction. **A game-theoretic perspective could be beneficially integrated with our proposed AI-side alignment framework to account for dynamic changes in human decision-making behavior**.
>
> (2) Relationship to Shapley value. The Shapley value is a solution concept in cooperative game theory that allocates the total payoff generated by a coalition fairly among its members in proportion to their marginal contributions. While the Shapley value has been successfully applied to fairness problems in machine learning contexts, **it does not naturally apply to our AI-assisted decision-making setting** due to fundamental structural differences. AI-assisted decision-making involves **no collective payoff requiring allocation**. Instead, each decision-maker operates independently on their own instances, deriving utility exclusively from their individual decisions. Moreover, the decision **utility is inherently non-transferable**, and the performance of one decision-maker cannot be redistributed to benefit or harm others.
>
> [1] E Bruce Goldstein. Cognitive psychology: Connecting mind, research, and everyday experience. Cengage learning Stamford, CT, 2015.
>
> [2] P.C. Cacciabue, F. Decortis, B. Drozdowicz, et al. A cognitive simulation model of human decision making and behavior in accident management of complex plants. IEEE Transactions on Systems, Man, and Cybernetics, 22(5):1058–1074, 1992.
>
> ### Comment & Question: [...] Do you think modifying AI confidence distributions could unintentionally change how different human groups rely on the AI?  [...]
>
> We thank you for this important question. Regarding confidence adjustments, we would like to clarify that **human trust in AI is primarily driven by experienced accuracy and outcome quality from AI- assistancenc ather than AI confidence values [1]**. Crucially, our approach improves fairness by **enhancing decision utility for disadvantaged groups** rather than degrading performance for advantaged groups, as demonstrated in `Appendix A.8.2 INSTANCE ACCURACY DISTRIBUTION ANALYSIS`. **When adjusted AI confidence is higher for a particular group, this genuinely reflects that such adjustments yield greater performance improvements for that group. Therefore, the confidence adjustment provides an informative signal that helps improve actual utility gains rather than a misleading factor that would distort trust.** `This holds true for both groups`, and thus we would like to clarify that `our approach is unlikely to cause differential reliance issues such as over-trusting or disengagement across different human groups.`
>
> [1] Yin, Ming, Jennifer Wortman Vaughan, and Hanna Wallach. "Understanding the effect of accuracy on trust in machine learning models." CHI. 2019.

---

> > ### Comment · Reviewer_wqhf · 2025-11-27
> > **Response**
> >
> > Thanks you so much for your consideration and checking my concerns.
> > I read your answers and it's enough for me now.
> > I have no more concerns at this time and I already gave you full accept.

---

### Author Response · Authors · 2025-11-21
**Updated Manuscript**

We kindly thank all the reviewers for their time and for providing valuable feedback on our work. We appreciate the recognition of our work as fresh (Reviewers MdV1, wqhf) and novel (Reviewer T784) , addressing an overlooked problem (Reviewers Wqv4, T784) with solid theoretical foundations (Reviewers wqhf, T784) and empirical credibility (Reviewers wqhf, T784, MdV1).

Based on the reviewers' suggestions, we have made the following revisions to our submission:

- `Modified Section 5.1 Settings - Hyperparameters`: added summary of parameter sensitivity analysis (Reviewer wqhf);
- `Modified Section 5.2 - Table 1`: added standard deviations and corrected bold highlighting (Reviewer Wqv4);
- `Added Appendix A.1`: detailed discussion on rationale and scope of Assumption 2.1, verification that Assumption 2.1 holds generally in real-world AI-assisted decision-making contexts, and validation of the robustness of our method under violations of Assumption 2.1 (Reviewer T784, Reviewer wqhf, Reviewer Wqv4);
- `Added Appendix A.8.1`: detailed quantitative ablation results (Reviewer wqhf);
- `Added Appendix A.8.2`: detailed instance decision accuracy distribution analysis, demonstrating fairness improvement by delivering better instance outcomes for the disadvantaged group without negatively impacting the advantaged group (Reviewer Wqv4);
- `Added Appendix A.8.5`: detailed computational complexity analysis, with theoretical and experimental validation of scalability with multi-attribute grouping (Reviewer T784);
- `Added Appendix A.8.6`: detailed extension to multi-class settings, with theoretical and experimental validation of scalability to multi-class scenarios (Reviewer MdV1, Reviewer wqhf);
- `Modified Appendix A.9.2`: detailed discussion on rationale and scope of archival dataset usage (Reviewer MdV1);
- `Added Appendix A.9.3`: discussion on connections to and distinctions from game-theoretic perspectives (Reviewer wqhf);
- `Added Appendix A.9.4`: discussion on potential Yule's effect and future integration with the stratification method (Reviewer Wqv4).

---

> ### Author Response · Authors · 2025-12-03
> **Rebuttal Summary**
>
> Dear Area Chair and Reviewers,
>
> We sincerely thank all reviewers for their thoughtful feedback. As the response deadline approaches, we would like to briefly summarize the status of our paper.
>
> > Paper summary
>
> Our work focuses on a previously overlooked problem: existing AI confidence calibrations fail to ensure fair utility across groups of decision-makers with heterogeneous cognitive capacities. We provide theoretical analysis establishing this problem's existence and propose a new AI confidence adjustment objective—inter-group alignment—which, combined with human alignment, theoretically bounds utility disparities. Building on this foundation, we propose cognition-aware multicalibration, an implementation that ensures interpretable utility fairness and optimal overall utility.
>
> > Discussion period summary
>
> We thank all reviewers for reviewing our paper carefully and offering constructive suggestions. During the discussion period,  **we have provided responses to all questions raised by the four reviewers**.
>
> **We conducted additional experiments and provided detailed discussions in both the rebuttal and the revised manuscript** to address the reviewers' all concerns, as listed in the above `Updated Manuscript`. We have received follow-up responses and acknowledgment from reviewers (**wqhf**, **Wqv4**, and **MdV1**).
>
> Sincerely,
>
> The Authors

---

### Meta-Review · Area_Chair_nDat · 2025-12-31

**Summary:**

All reviewers liked the paper, although some with a low confidence. (The lowest initial score of 6 has been improved to 8 during the discussion period by one of the reviewers.) Both of reviewers with initial score of 8.0 seem to have utilized gen-AI (cf. https://iclr.pangram.com/reviews?submission_number=14869).

**Reviewer Concerns:**

The authors have made a substantial revision during the rebuttal phase. Still, there are multiple multiple concerns outstanding:

- evaluating cognitive disparities and using them in decision making might be unethical.

- monotonicity assumptions (Assumption 2.1) are rather strong and this should be stressed in the main body of the paper.

- empirical evaluation is somewhat limited. The authors test on a single dataset (https://dl.acm.org/doi/abs/10.1145/3514094.3534150, AIES '22).

**Reviewer Scores:**

Reviewers wqhf and T784 suggested a score of 8.0. Wqv4 suggested he would incrase the score to 8. Reviewer MdV1 intended to keep his score of 6.0.

---

### Decision · Program_Chairs · 2026-01-26

Accept (Poster)